# Grokking at the Edge of Linear Separability

**Alon Beck** [1]  **Noam Levi** [2]  **Yohai Bar Sinai** [1]

## Abstract

We investigate the phenomenon of grokking – delayed generalization accompanied by non-monotonic test loss behavior – in a simple binary logistic classification task, for which "memorizing" and "generalizing" solutions can be strictly defined. Surprisingly, we find that grokking arises naturally even in this minimal model when the parameters of the problem are close to a critical point, and provide both empirical and analytical insights into its mechanism. Concretely, by appealing to the implicit bias of gradient descent, we show that logistic regression can exhibit grokking when the training dataset is nearly linearly separable from the origin and there is strong noise in the perpendicular directions. The underlying reason is that near the critical point, "flat" directions in the loss landscape with nearly zero gradient cause training dynamics to linger for arbitrarily long times near quasi-stable solutions before eventually reaching the global minimum. Finally, we highlight similarities between our findings and the recent literature, strengthening the conjecture that grokking generally occurs in proximity to the interpolation threshold, reminiscent of critical phenomena often observed in physical systems.

## 1. Introduction

Understanding the relationship between the intrinsic properties of data, the training dynamics of neural networks (NNs), and their ability to generalize is crucial to explaining the success of modern machine learning (ML) algorithms. In particular, highly over-parameterized models based on the transformer architecture (Vaswani et al., 2023), such as

[1]Raymond and Beverly Sackler School of Physics and Astronomy, Tel Aviv University, Tel Aviv 69978, Israel [2]École Polytechnique Fédérale de Lausanne (EPFL), Lausanne, Switzerland. Correspondence to: Alon Beck <alonbk2@gmail.com>, Noam Levi <noam.levi@epfl.ch>.

*Proceedings of the 42nd International Conference on Machine Learning*, Vancouver, Canada. PMLR 267, 2025. Copyright 2025 by the author(s).

Large Language Models (LLMs) (OpenAI, 2024; Google, 2023; Zeng et al., 2022; Brown et al., 2020; Chowdhery et al., 2022; Anil et al., 2023), as well as state of the art models for computer vision (Srivastava & Sharma, 2023), defy expectations and are able to generalize with a number of parameters far exceeding the so called interpolation threshold (Kaplan et al., 2020; Schaeffer et al., 2023). Interestingly, these models have been shown to exhibit unpredictable behaviors when changing the number of network parameters, not only with respect to generalization, but also in their learning dynamics.

One such phenomenon, known as Grokking, was first observed by Power et al. (2022) during the training of a transformer model on modular arithmetic tasks. Grokking occurs when a model initially achieves perfect training accuracy but no generalization (i.e. no better than a random predictor), and upon further training, transitions to almost perfect generalization. This phenomenon has garnered substantial attention in recent years (Gromov, 2023; Liu et al., 2023; Xu et al., 2023) due to its striking contrast with naive expectations, whereby over-fitting is generally seen as an undesirable property of models that should not generalize with further training, originally dealt with using early stopping (Prechelt, 1996).

In this work, we present a straightforward model where grokking naturally emerges, allowing us to identify and analyze its fundamental cause as proximity of the system to a critical point. In our simple yet illuminating setting, the asymptotic optimal solution can always be identified, allowing a sharp definition of notions that are typically ambiguous, such as "memorization" and "learning".

We study a typical logistic binary classification problem, with the goal of finding a linear separator between two Gaussians with distinct labels. We assume that the Gaussians are well separated along the separation axis and contains noise in all perpendicular directions. Extensions of this setup are considered in Sec. 5. We mainly focus on the limit of large $N, d \to \infty$ while keeping the ratio $\lambda = d/N$ fixed, although the results can be generalized.

Our main contributions are:

1. We prove, and demonstrate numerically, that grokking may occur in this setting, and is promoted when $\lambda$ is close to 1/2 and the separation along the separation

between the Gaussians is small relative to the noise in other directions.

2. We show that this happens because $\lambda = 1/2$ is a *critical point*. That is, for $\lambda < 1/2$ the model will almost surely asymptotically approach perfect generalization accuracy and vanishing loss, while for $\lambda > 1/2$ the model will almost surely achieve imperfect generalization accuracy and the population loss will diverge at $t \to \infty$.

3. More fundamentally, we show that generalization depends on whether the training set is *linearly separable* from the origin (that is, whether the origin is contained in the convex hull of the training set). In the limit $N, d \to \infty$, the data almost surely separable from the origin if $\lambda > 1/2$ and almost surely false otherwise.

4. Moreover, we show that near the threshold value $\lambda = 1/2$ (or, more generally, when the data is on the verge of being separable), the dynamics may generically track the overfitting solution for arbitrarily long times before transitioning to the optimal generalizing solution. This behavior manifests as a non-monotonic test loss and delayed generalization, and can lead to divergence of the "grokking time".

5. We construct a simple, one-dimensional model which captures the salient aspects of the problem, and explicitly solve the time evolution of the model parameters for several interesting cases.

The main takeaway from this setup is that *grokking happens near a critical point*, similar to "critical slowing down" in the physics literature. While further study is needed, we conjecture that this applies to other grokking examples, as was demonstrated explicitly (though not necessarily stated in these terms) in Levi et al. (2023); Liu et al. (2023); Gromov (2023); Rubin et al. (2023; 2024).

The rest of the paper is organized as follows: Sec. 3 presents our main analysis of grokking as a critical phenomenon, beginning with empirical results in Sec. 3.1, studying the possible solutions in Sec. 3.2, and relating it to linear separability in Sec. 3.3. Finally, we bring together the pieces in Sec. 3.4 to explain why grokking occurs near the critical point. Sec. 4 provides a tractable effective model which fully captures the grokking dynamics. In Sec. 5, we discuss generalizations of this setup. We conclude in Sec. 6 and discuss future directions. In the appendices we discuss several generalizations of our results and provide some further proofs and derivations.

## 2. Related Work

Following the discovery of grokking by Power et al. (2022), numerous studies have attempted to elucidate its underlying mechanisms. Liu et al. (2022) showed that when sufficient data determines the structured representation, perfect generalization can be achieved on a non-modular addition task. Other works have identified factors contributing to grokking, including pattern learning (Davies et al., 2023), delayed robustness (Humayun et al., 2024), and transitions from memorization to circuit formation (Nanda et al., 2023), and the role of activation sparsity, weight entropy, and circuit complexity in real-world tasks (Golechha, 2024). Others analyzed the trigonometric algorithms learned by networks after grokking (Nanda et al., 2023; Chughtai et al., 2023; Merrill et al., 2023; Gromov, 2023), and demonstrated similar dynamics in sparse parity tasks (Merrill et al., 2023). Additional works proposed "slingshots" (Thilak et al., 2022) or "oscillations" (Notsawo et al., 2023) as explanations for grokking, whereas others focused on the role of regularization (Power et al., 2022; Liu et al., 2023) and of numerical precision (Prieto et al., 2025), which may significantly impact grokking in certain scenarios. We stress that our work requires **none** of these in order to exhibit or explain grokking.

Recently, a body of works on solvable models which grok in various settings has emerged. Liu et al. (2023); Kumar et al. (2023) and Lyu et al. (2024) have linked grokking to memorization and transitions from lazy to rich dynamics. Žunkovič & Ilievski (2022); Gromov (2023); Doshi et al. (2024) analyzed solvable models exhibiting grokking and related their findings to the formation of latent-space structure, Xu et al. (2023) related grokking to benign over-fitting for ReLU networks on XOR data, Rubin et al. (2024) described grokking as a first-order phase transition, and Levi et al. (2023) provide the full dynamical solution of grokking in linear regression. Our work attempts to sidestep external probes, and fill the gap between solvable models and representation learning, whereby we can always identify the optimal solutions, while still solving the dynamics of the model. To accomplish this, our work relies on the results of Soudry et al. (2018); Nacson et al. (2019); Ji & Telgarsky (2019), which analyze the late-time dynamics properties of logistic regression for separable and inseparable data under gradient descent (GD).

## 3. Grokking in Binary Classification

### 3.1. Model setup and empirical results

Consider a dataset of $N$ training samples $\{\tilde{\boldsymbol{x}}_i\}_{i=1}^N \subset \mathbb{R}^{d+1}$. We investigate the case where the data consists of two Gaussian distributions with opposite labels, well separated along the axis of separation but with added noise in all

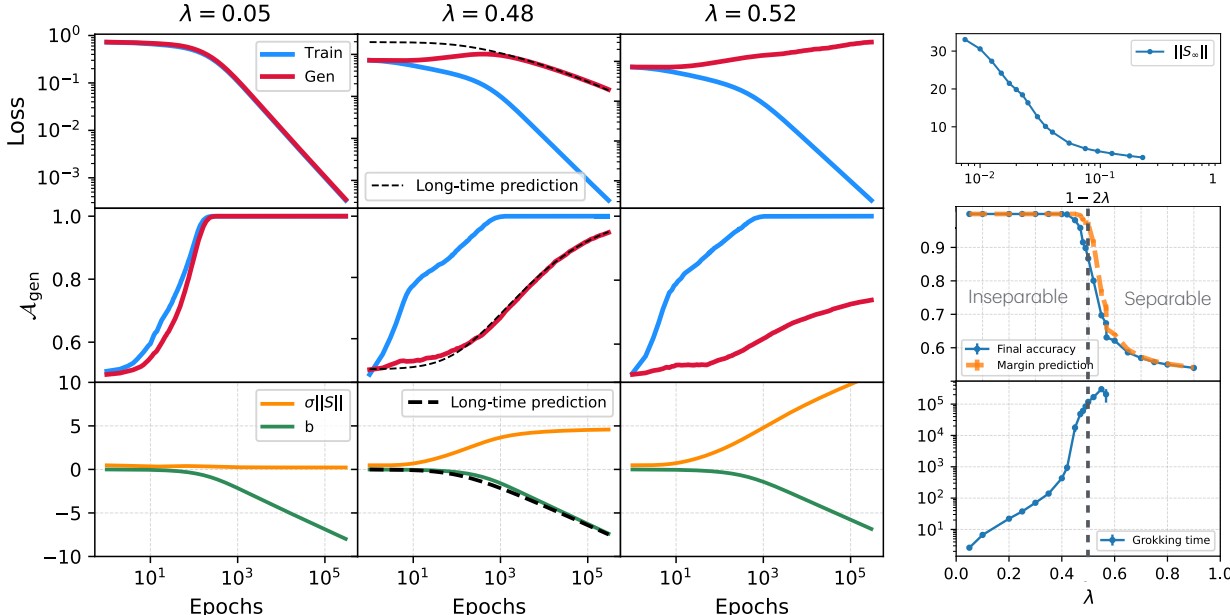

*Figure 1.* **Left panels: Gradient descent dynamics for three different values of** $\lambda = d/N$**.** Loss and accuracy over the train and test datasets, and the time evolution of $b(t)$ and $\|\boldsymbol{S}(t)\|$. Grokking is significant only when $\lambda$ approaches to $1/2$ from below. We can see that for $\lambda > 1/2$, $\|\boldsymbol{S}\|$ increases indefinitely and generalization is not possible (see Eq. (4)). The parameters are $N = 4 \cdot 10^4$, $\sigma = 5$, $\eta = 0.01$. The direction of $\boldsymbol{S}(t=0)$ was drawn isotropically with $\|\boldsymbol{S}_0\| = 0.1$ and $b(t=0) = 0$. The number of test samples is $N_{\text{test}} = 10^4$. **Right panels**: Top: The norm of the limiting value $\boldsymbol{S}_\infty$ in the separable case $\lambda > 1/2$, as a function of $\lambda$. Note that $\|\boldsymbol{S}_\infty\|$ diverges for $\lambda \to \frac{1}{2}$. Middle: the accuracy at the end of the training (in blue), and the predicted limiting accuracy (orange), calculated only using the margin of the dataset, see Prop. 3.2. Bottom: The Grokking time, defined here as the delay between the times when $\mathcal{A}_{\text{train/gen}}$ surpass a threshold of 0.9. Grokking time and $\|\boldsymbol{S}_\infty\|$ diverge near $\lambda = 1/2$. Additional details regarding the experiments can be found in App. K.

other directions. WLOG, we take the separation between the Gaussians to be along the first axis. Explicitly, the distribution of the data is $\tilde{\boldsymbol{x}}_i \sim \mathcal{N}(\pm\boldsymbol{\mu}, \Sigma)$ where the covariance $\Sigma$ is diagnoal with $\Sigma_{11} = \sigma_A$ and $\Sigma_{ii} = \sigma_B$ for $i > 1$, and $\boldsymbol{\mu} = (\mu, 0, 0, ...)$ is a vector pointing in the direction of the first axis with magnitude $\mu$. We consider a binary logistic regression task with a linear model without bias.

Throughout this work, we focus on the regime $\sigma_A \ll \mu$, which allows full generalization below the critical point. We explain the reasoning behind this assumption, and discuss cases where it does not hold in Sec. 5. We also note that neither Gaussianity nor the assumption of unit covariance is essential for our analysis, and are only made for simplicity (see App. G).

**Reduction to a $d$-dimensional problem with bias.**

We consider now the limiting case $\sigma_A = 0$, in which the first coordinate of each data point is simply $\pm\mu$. In this case, the gradient flow dynamics can be exactly mapped to a $d$-dimensional problem with bias, in which all points are assigned the *same label*. The key idea is that, up to a scaling factor, the data points take the form $\tilde{\boldsymbol{x}}_i = (1, \boldsymbol{x}_i)$, where $\boldsymbol{x}_i \sim \mathcal{N}(0, \frac{\sigma_B^2}{\mu^2}\boldsymbol{I}_d)$ is a d-dimensional vector ($\boldsymbol{I}_d$ is the $d \times d$

identity matrix). In this formulation, it is standard to treat the first coordinate of the weight vector as the bias term. A more rigorous justification for this equivalence is provided in App. H, where it is also shown that under this mapping the $d$-dimensional problem the effective learning rate is faster by a factor of $\mu^2$. The benefit of this equivalence is that the analysis of the equivalent model is simpler.

*To sum up, for simplicity, throughout this paper we will analyze the following equivalent d-dimensional problem:* Consider $N$ training samples $\{\boldsymbol{x}_i\}_{i=1}^N \subset \mathbb{R}^d$ where $\boldsymbol{x}_i \sim \mathcal{N}(0, \sigma^2\boldsymbol{I}_d)$ and $\sigma > 0$ is the feature standard deviation (mapped to $\sigma_B/\mu$ in the model above). Consider logistic classification, where all input points are assigned the same label, which for concreteness we take as $\{y_i\}_{i=1}^N = -1$.

**Linear model: loss and accuracy.** The model parameters are a weight vector $\boldsymbol{S} \in \mathbb{R}^d$ and a bias term $b \in \mathbb{R}$. The output is a scalar $f_i = f(\boldsymbol{x}_i) = \boldsymbol{S} \cdot \boldsymbol{x}_i + b$. We optimize the empirical cross-entropy loss $\mathcal{L}(\boldsymbol{S}, b)$ and measure the

empirical accuracy $\mathcal{A}(\boldsymbol{S}, b)$, given by[1]

$$\mathcal{L}(\boldsymbol{S}, b) = \frac{1}{N} \sum_{i=1}^{N} \ell\left(\boldsymbol{S}^T \boldsymbol{x}_i + b\right) , \qquad (1)$$

$$\mathcal{A} = \frac{1}{N} \sum_{i=1}^{N} \Theta\left(-\boldsymbol{S}^T \boldsymbol{x}_i - b\right) ,$$

where $\ell(f_i) = \log\left(1 + e^{-y_i \cdot f_i}\right) = \log\left(1 + e^{f_i}\right)$ is the single sample loss and $\Theta(z)$ is the Heaviside function, defined as $\Theta(z) = 1$ if $z \geq 0$ and $\Theta(z) = 0$ if $z < 0$.

**Optimizer.** Throughout the main text we use gradient descent (GD) dynamics. The effects of other optimizers are discussed in App. F. The GD equations at training step $t$ with learning rate $\eta$ are

$$\boldsymbol{S}_{t+1} - \boldsymbol{S}_t = -\eta \nabla_{\boldsymbol{S}} \mathcal{L}, \qquad b_{t+1} - b_t = -\eta \partial_b \mathcal{L}. \quad (2)$$

In this paper we will focus on the gradient flow (GF) limit $(\eta \to 0)$ of these equations.

**Numerical results.** In Fig. 1, we show numerical results depicting the gradient-descent dynamics of the model across three values of $\lambda \equiv d/N$. Notably, we observe a significant grokking effect, both in the non-monotonicity of the test loss, and a delayed rise in test accuracy, only when $\lambda \to \lambda_c = 1/2$ (there may be some differences between the grokking observed here and other examples in the literature, but they seem to be superficial - see App. B). In the following section, we explain how $\lambda_c$ can be interpreted as the interpolation threshold in this setting.

### 3.2. The generalizing and over-fitting solutions

To understand grokking in this setup, we begin by examining the optimal generalizing solution. Since the support of the input distribution is unbounded and all labels are equal, the model must position all points in $\mathbb{R}^d$ on the same side of the separating hyperplane, effectively pushing the decision boundary to infinity.

To see this rigorously, we derive expressions for the generalization accuracy and loss. Since the data follows a Gaussian distribution, $\boldsymbol{x}_i \sim \mathcal{N}(0, \sigma^2 \boldsymbol{I}_d)$, the generalization (population) loss is, by definition:

$$\mathcal{L}_{\text{gen}} = \mathbb{E}_{\boldsymbol{x} \sim \mathcal{N}(0, \sigma^2 \boldsymbol{I}_d)} \left[\log\left(1 + \exp\left(\boldsymbol{S}^T \boldsymbol{x} + b\right)\right)\right] \quad (3)$$

$$= \mathbb{E}_{y \sim \mathcal{N}(0,1)} \left[\log\left(1 + e^{\sigma \|\boldsymbol{S}\| y + b}\right)\right] ,$$

where we used the fact that $\boldsymbol{S}^T \boldsymbol{x} \sim \mathcal{N}(0, \|\boldsymbol{S}\|^2 \sigma^2)$. Note that $\mathcal{L}_{\text{gen}}$ depends only on $b$ and $\|\boldsymbol{S}\|$. Similarly, the

generalization accuracy is given by:

$$\mathcal{A}_{\text{gen}}(\boldsymbol{S}, b) = \mathbb{E}_{y \sim \mathcal{N}(0,1)} \left[\Theta\left(-\sigma \|\boldsymbol{S}\| y - b\right)\right] \quad (4)$$

$$= \frac{1}{2} \left[1 - \text{erf}\left(\frac{1}{\sqrt{2}} \frac{b}{\sigma \|\boldsymbol{S}\|}\right)\right] ,$$

where $\text{erf}$ is the error function.

**Proposition 3.1.** *Perfect generalization, i.e., $\mathcal{L}_{\text{gen}} \to 0$ and $\mathcal{A}_{\text{gen}} \to 1$, is achieved only if both $b \to -\infty$ and $b/\|\boldsymbol{S}\| \to -\infty$. That is, $b$ must tend to negative infinity while also being infinitely large compared to $\|\boldsymbol{S}\|$.*

*Proof.* It is easily seen from Eq. (4) that the condition $\mathcal{A}_{\text{gen}} \to 1$ requires $b/\|\boldsymbol{S}\| \to -\infty$. If $b$ is bounded, then this can only happen for $\|\boldsymbol{S}\| \to 0$, but this cannot be since then Eq. (3) implies that $\mathcal{L}_{\text{gen}}$ is bounded away from zero. Therefore, perfect generalization implies both $b \to -\infty$ and $b/\|\boldsymbol{S}\| \to -\infty$. $\qquad\square$

The bottom panels of Fig. 1 show that $b \to -\infty$ at late times in all parameter regimes. However, while $\|\boldsymbol{S}\|$ saturates at a constant value for $\lambda < 1/2$, it diverges when $t \to \infty$ for $\lambda > 1/2$, and does so at a rate comparable to $b$, leading to sub-optimal generalization $\lim_{t \to \infty} \mathcal{A}(\boldsymbol{S}(t), b(t)) < 1$.

**Relation to prior results regarding separability.** These results are closely related to the framework developed by Soudry et al. (2018), who studied the convergence of binary classification for linearly separable data, and later expanded by Ji & Telgarsky (2019) for inseparable data. In our case, since the model contains a bias term and all labels are the same, the data is always separable by a hyperplane "at infinity". To use their framework, we need to work in an extended space of dimension $d + 1$, where we define the extended weight vector $\boldsymbol{w} = (\boldsymbol{S}, b) \in \mathbb{R}^{d+1}$. The network solution at the late stages of training can be obtained as a direct corollary of Theorem 3 from Soudry et al. (2018).

**Theorem 1** (Rephrased from Theorem 3 of Soudry et al. (2018) ). *In the setting described above, for any smooth monotonically decreasing loss function with an exponential tail, and for small learning rate, GD iterates will converge at the late stages of training to:*

$$\boldsymbol{w}(t) = \boldsymbol{w}_{\text{SVM}} \log(t) + \boldsymbol{\rho}(t) , \qquad (5)$$

$$\boldsymbol{w}_{\text{SVM}} = \underset{(\boldsymbol{S}, b)}{\operatorname{argmin}} \left\{\|\boldsymbol{S}\|^2 + b^2 \quad s.t. \quad \boldsymbol{S}^T \boldsymbol{x}_i + b \leq -1\right\}.$$

*Here, $\boldsymbol{w}_{\text{SVM}}$ is the solution[2] to the hard margin SVM problem in the extended $d + 1$ space and $\boldsymbol{\rho}$ is a residual vector which is bounded for all $t$.*

Connecting this result to the previous discussion, we see

---

[1] The labels do not appear explicitly in $\mathcal{L}$ since they are identical for all samples.

[2] Note that the SVM solution in the extended $d + 1$ is not the same as the typical formulation of the Support Vector Machine (SVM) with bias in $d$ dimensions, because of the different penalty used for the bias term.

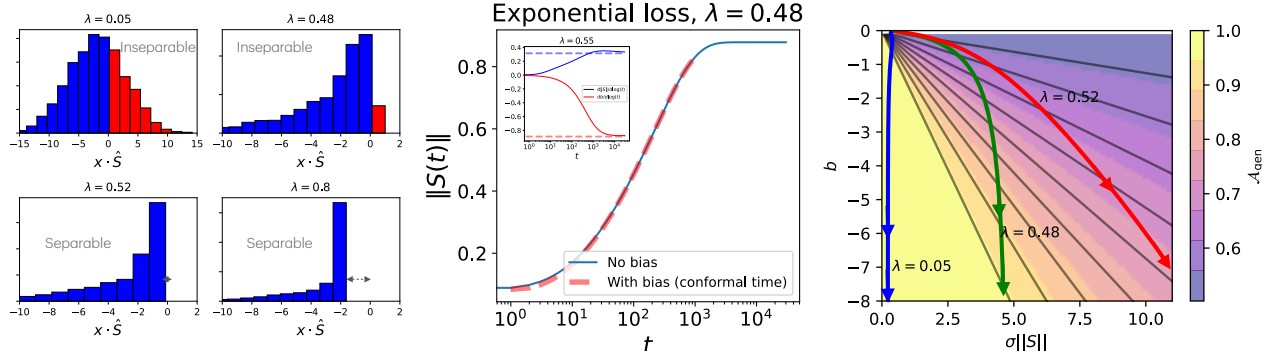

*Figure 2.* **Evolution of the model parameters.** (left) the distribution of $\boldsymbol{S}^T x_i/\|\boldsymbol{S}\|$, where $\boldsymbol{S}$ is the final spatial weight vector that was found using GD dynamics for $\lambda = 0.05, 0.48, 0.52, 0.8$. The parameters are identical to those of Fig. 1. We can see that for $\lambda = 0.05, 0.48$ the model does not separate the data (because the data is inseparable) while for $0.52, 0.8$ it does. The margin is plotted for $\lambda = 0.8$ Middle panel: $\|\boldsymbol{S}(t)\|$, optimized with GD using the exponential loss given in Eq. (6), with and without a bias term. With a bias term, the result is shown as a function of the conformal time (Eq. (8)). The two curves follow the same path different rates. The inset shows $\frac{d\|\boldsymbol{S}\|}{d\log(t)}$ and $\frac{db}{d\log(t)}$, (right) Optimization paths for different $\lambda$ values, shown in the $b, \sigma\|\boldsymbol{S}\|$ plane. For inseparable data $b$ diverges while $\boldsymbol{S}$ is bounded, while slightly above the limit of separability both $b$ and $\|\boldsymbol{S}\|$ diverge.

indeed that either $|b|$ and/or $\|\boldsymbol{S}\|$ must diverge at infinite training times, and the question is now reduced to the directionality of $\boldsymbol{w}_{\mathrm{SVM}}$.

The generalizing solution, which classifies correctly all points in $\mathbb{R}^d$ is when $\boldsymbol{w}_{\mathrm{SVM}} = (\boldsymbol{0}, -1)$, ($\boldsymbol{0}$ being the $d$-dimensional zero vector) i.e. when it points in the direction of the bias and the separating plane is at infinity. This is exactly the aforementioned condition $b \to -\infty$ and $|b|/\|\boldsymbol{S}\| \to \infty$. In contrast, over-fitting occurs when the hyperplane is far enough from the data to correctly classify all the training samples, but does not go to infinity. In the extended space, this means that $\boldsymbol{w}_{\mathrm{SVM}}$ also contains a component in the direction of the data, and the model did not correctly learn the data distribution. In what follows, we will show that the factor determining whether we observe grokking in this setup is not the regular separability of data points from one another, but rather "separability from the origin" (or, separability with no bias), defined as follows:

**Definition 2.** *A data-set $\{x_i\}_{i=1}^N$, $x_i \in \mathbb{R}^{d \times 1}$ is linearly separable from the origin iff there exists a vector $\boldsymbol{S} \in \mathbb{R}^{d \times 1}$ such that $\boldsymbol{S}^T x_i > 0$ for any $i$.*

In the rest of the paper, we will use "separable" as a shorthand for "separable from the origin". We are now ready to present our main claims regarding the grokking phenomenology presented in Fig. 1. We argue that:

- The generalization and overfitting at $t \to \infty$ depend only on whether the training samples (in $\mathbb{R}^d$) are separable (from the origin) (Prop. 3.2).

- For large $N, d$, the training set is separable if $\lambda > \frac{1}{2}$ and inseparable for $\lambda < \frac{1}{2}$ (Prop. 3.3).

- For separable training sets ($\lambda > \frac{1}{2}$), the model will always overfit, and the limiting generalization accuracy is directly related to the optimal separating

margin (Prop. 3.2.2). For inseparable training sets ($\lambda < \frac{1}{2}$) the model will always generalize perfectly: $\lim_{t\to\infty} b(t) = \infty$ and $\boldsymbol{S}$ saturates on a finite value $\lim_{t\to\infty} \boldsymbol{S}(t) = \boldsymbol{S}_\infty$ (Prop. 3.2.1).

- However, for $\lambda \to \frac{1}{2}^-$, the training set is on the verge of separability, and $\|\boldsymbol{S}_\infty\|$ diverges (Prop. 3.4).

- Consequently, our main result follows: dynamics may take arbitrarily long times to reach the generalizing solution. This is the underlying mechanism of grokking in this setting.

### 3.3. Separability determines whether the model will generalize perfectly or not

**Proposition 3.2.** *The model will reach perfect generalization if and only if the data is not linearly separable from the origin. In particular:*

1. *If the data is not linearly separable from the origin, then $\lim_{t\to\infty} b(t) = \infty$ while $\boldsymbol{S}$ saturates on a finite value $\lim_{t\to\infty} \boldsymbol{S}(t) = \boldsymbol{S}_\infty$.*
2. *If the data is linearly separable from the origin, then $\lim_{t\to\infty} \mathcal{A}_{\mathrm{gen}} = \frac{1}{2}\left[1 + \mathrm{erf}\left(\frac{1}{\sigma M \sqrt{2}}\right)\right]$, where $M$ is the margin.*

To prove Prop. 3.2, we first note that due to the "exponential tail" of the cross-entropy loss, at late times the loss is dominated by samples with large model outputs $f = \boldsymbol{S}^T \boldsymbol{x} + b$, for which the cross-entropy loss $\ell(f)$ of Eq. (2), approaches the exponential loss $\ell_e(f) = e^f$ (Soudry et al., 2018; Nacson et al., 2019; Ji & Telgarsky, 2019). Specifically, the exponential loss must converge to the same late time dynamics as the cross entropy loss. Therefore, we will consider the exponential loss for which the calculations

are tractable,

$$\mathcal{L}_e(\boldsymbol{S}, b) = \frac{1}{N} \sum_{i=1}^{N} e^{\boldsymbol{S}^T \boldsymbol{x}_i + b}. \tag{6}$$

In the gradient-flow limit, the induced dynamics are

$$\frac{\partial \boldsymbol{S}}{\partial t} = -\frac{\eta}{N} e^b \sum_i e^{\boldsymbol{S}^T \boldsymbol{x}_i} \boldsymbol{x}_i, \quad \frac{\partial b}{\partial t} = -\frac{\eta}{N} e^b \sum_i e^{\boldsymbol{S}^T \boldsymbol{x}_i}. \tag{7}$$

Note that both rates are proportional to a common time-dependent scalar $e^{b(t)}$. We can thus define a so-called *conformal time*[3]: $\tau(t) = \int_0^t e^{b(t')} dt'$, which is a strictly increasing function of $t$. In terms of $\tau$, the time evolution takes the form

$$\frac{\partial \boldsymbol{S}}{\partial \tau} = -\frac{\eta}{N} \sum_i e^{\boldsymbol{S}^T \boldsymbol{x}_i} \boldsymbol{x}_i, \quad \frac{\partial b}{\partial \tau} = -\frac{\eta}{N} \sum_i e^{\boldsymbol{S}^T \boldsymbol{x}_i}. \tag{8}$$

The importance of this change of variables is that the dynamics of $\boldsymbol{S}(\tau)$ in terms of the conformal time are identical to those of $\boldsymbol{S}(t)$ *in the absence of bias*. That is, $\boldsymbol{S}(t)$ follows the same path that would be obtained by minimizing $\mathcal{L} = \frac{1}{N} \sum_{i=1}^{N} e^{\boldsymbol{S}^T \boldsymbol{x}_i}$, but does so at a different rate which depends exponentially on the current value of $b(t)$. This is demonstrated in the middle panel of Fig. 2. Since $\tau(t)$ diverges for $t \to \infty$ (see App. C for details), $\boldsymbol{S}(t)$ must follow the same path at long times, as it would have followed without bias. We can now complete the proof:

**Proof of 3.2.1 (inseparable case)** Consider the dynamics *without* bias. In case the data is inseparable, $\mathcal{L}_e$ of Eq. (6) is unbounded in all directions of $\boldsymbol{S}$. That is, for any unit vector $\boldsymbol{e} \in \mathbb{R}^d$ we have $\lim_{\alpha \to \infty} \mathcal{L}_e(\alpha \boldsymbol{e}, 0) = \infty$. Since $\mathcal{L}_e$ is convex, the gradient flow dynamics will lead to a global minimum at a finite point $\lim_{t \to \infty} \boldsymbol{S}(t) = \boldsymbol{S}_\infty$. Recalling the discussion of conformal time above, this is also the limit of the dynamics *with* bias. From Eq. (5), we know that either $\|\boldsymbol{S}(t)\|$ or $|b(t)|$ must diverge, and since $\boldsymbol{S}(t)$ approaches a finite value, we conclude that $|b|/\|\boldsymbol{S}\| \to \infty$ for $t \to \infty$. That is, $\boldsymbol{w}_{\mathrm{SVM}} = (0, -1)$ and the model flows towards the generalizing solution.

**Proof of 3.2.2 (separable case)** When the training set is separable from the origin, it is easier to examine the optimization problem in Eq. (5) directly. We wish to minimize $\|\boldsymbol{w}\|^2 = \|\boldsymbol{S}\|^2 + b^2$ under the separability constraints. The generalizing solution $\boldsymbol{w}_g = (0, -1)$ satisfies all constraints trivially and has $\|\boldsymbol{w}_g\| = 1$. However, since the data is separable from the origin, there exists another solution to the constraints, namely $\boldsymbol{w}^* = (\boldsymbol{S}^*, 0)$, where $\boldsymbol{S}^*$ is the separating vector in $d$ dimensions

---

[3]This is a common measure in cosmology and gravitational physics to describe co-moving objects in an expanding or shrinking spacetime background (Guth, 1981).

without bias, i.e. the solution to

$$\boldsymbol{S}^* = \operatorname*{argmin}_{\boldsymbol{S}} \left\{ \|\boldsymbol{S}\|^2 \quad \text{s.t.} \quad \boldsymbol{S}^T \boldsymbol{x}_i \leq -1 \right\}. \tag{9}$$

The norm of $\boldsymbol{S}^*$ is the inverse of the separation margin $M = 1/\|\boldsymbol{S}^*\|$.

Due to convexity, any convex combination of $\boldsymbol{w}_g$ and $\boldsymbol{w}^*$ will also satisfy the constraints, and since they are orthogonal it also has a smaller norm. The combination with the smallest norm is the global optimum, which is easily shown to be proportional to $\boldsymbol{w}_{\mathrm{SVM}} \propto (M^2 \boldsymbol{S}^*, -1)$. That is, both $\boldsymbol{S}$ and $b$ diverge when $t \to \infty$ and $\lim_{t \to \infty} \frac{b(t)}{\|\boldsymbol{S}(t)\|} = -\frac{1}{M}$. Plugging the result into Eq. (4) completes the proof. Next, we establish the relation between separability and $\lambda$.

**Proposition 3.3.** *For $N, d \to \infty$, and $\lambda = d/N$, the dataset is separable from the origin with probability $1$, if $\lambda > 1/2$, and is inseparable if $\lambda < 1/2$.*

In other words, Prop. 3.3 states that (almost) any large set of $N$ points in $d$ dimensions are separable (i.e., lie on the same "half" of some hypersphere passing through origin) as long as $d > N/2$. This is a direct corollary of Wendel's theorem (Wendel, 1962) which we prove in App. A.

### 3.4. Collecting the pieces: why does grokking happen near $\lambda = \frac{1}{2}$?

We have established that for $\lambda < \frac{1}{2}$, the model will almost surely generalize perfectly. For infinitely long times, $\boldsymbol{S}(t)$ converges to a finite vector $\boldsymbol{S}_\infty$, and $b(t)$ diverges. For $\lambda > \frac{1}{2}$, the model will almost surely overfit. Intuitively, one should expect that in the vicinity of the critical point $\lambda_c = \frac{1}{2}$, where the two solutions exchange stability, dynamics may become slow. This is because for $\lambda > 1/2$ the overfitting solution is stable and for $\lambda$ smaller than but close to $1/2$, it is unstable but only slightly so. Therefore, the dynamics may spend arbitrarily long times in the vicinity of the overfitting solution before flowing to the generalizing solution. This is delayed generalization. Rigorously, this happens through of the following properties:

**Proposition 3.4.** *For $\lambda \to \frac{1}{2}^-$, $\|\boldsymbol{S}_\infty\| \to \infty$.*

That is, when the training set is non-separable, but on the verge of separability, $\|\boldsymbol{S}_\infty\|$ obtains arbitrarily large values. This statement is formally proven in App. D and empirically demonstrated in Fig. 1. It can also be obtained as a corollary of Ji & Telgarsky (2019). An intuitive geometric interpretation is that for a nearly separable set, $\boldsymbol{S}(t)$ approaches a finite limit $\boldsymbol{S}_\infty$, but a small translation of the data would make the set separable, and correspondingly would make $|\boldsymbol{S}(t)| \to \infty$. Smoothness thus implies $|\boldsymbol{S}_\infty|$ must be large if the set is almost separable.

**Proposition 3.5.** *For $\lambda < \frac{1}{2}$ and $\sigma$ large enough, $\boldsymbol{S}(t)$ will approach its asymptotic value $\boldsymbol{S}_\infty$ arbitrarily fast.*

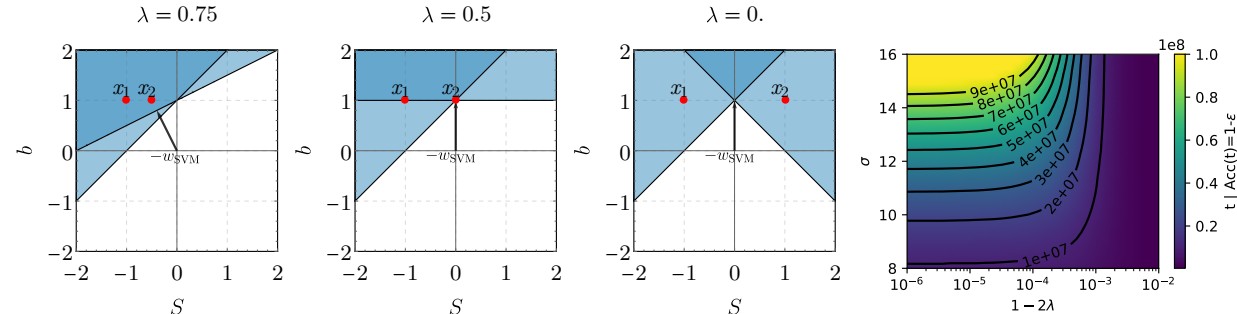

*Figure 3.* **Simplified model.** Three left panels: Illustration of the hard margin SVM problem in 1+1 dimensions for the simplified model. Note that $-w_{SVM}$ is the point closest to the origin in the intersection of the two shaded regions. When $w_{SVM}$ points along the bias axis (the vertical axis) if and only if $x_2 \geq 0$. (right) Grokking time, defined as the time it takes for the generalization accuracy to reach 0.95, plotted against $\sigma$ and $1 - 2\lambda$. We see it diverges when both $\lambda \to 1/2$ and $\sigma \to \infty$, while neither condition suffices alone.

|  | $\lambda < 1/2\,(x_2 > 0)$ | $\lambda = 1/2\,(x_2 = 0)$ | $\lambda > 1/2\,(x_2 < 0)$ |
|---|---|---|---|
| $b(t \gg 1)$ | $-\log(t)$ | $-\log(t)$ | $-\frac{1}{1+M^2}\log(t)$ |
| $\|\boldsymbol{S}\|(t \gg 1)$ | $\frac{1}{2(1-\lambda)}\log\left(\frac{1}{1-2\lambda}\right)$ | $\log(\log(t))$ | $\frac{M}{1+M^2}\log(t)$ |

*Table 1.* Summary of the different regimes of the simplified model, where $x_1 = -1$, $x_2 = 1 - 2\lambda$, and the margin is $M = |x_2| = 2\lambda - 1$.

*Proof.* To see this, it is useful to define the rescaled variables $\tilde{\boldsymbol{x}}_i = \boldsymbol{x}_i/\sigma$, $\tilde{\boldsymbol{S}} = \sigma\boldsymbol{S}$. Clearly, $\tilde{\boldsymbol{x}}_i \sim \mathcal{N}(0, \boldsymbol{I_d})$. The gradient flow equations in terms of the rescaled variables are (we study the exponential loss for simplicity)

$$\frac{\partial \tilde{\boldsymbol{S}}}{\partial t} = -\sigma^2 \frac{\eta}{N} e^b \sum_i e^{\tilde{\boldsymbol{S}}^T \tilde{\boldsymbol{x}}_i} \tilde{\boldsymbol{x}}_i, \qquad (10)$$

$$\frac{\partial b}{\partial t} = -\frac{\eta}{N} e^b \sum_i e^{\tilde{\boldsymbol{S}}^T \tilde{\boldsymbol{x}}_i}.$$

Note that these are identical to the gradient flow equations of the original variables given by Eq. (7), but the dynamics of $\boldsymbol{S}$ are *faster* by factor of $\sigma^2$. Thus, by taking a large $\sigma$, $\tilde{\boldsymbol{S}}$ will approach its asymptotic value $\boldsymbol{S}_\infty$ arbitrarily fast, while the dynamics of $b$ will not change. Recalling that $\sigma$ was mapped to $\sigma_B/\mu$ in the original two-Gaussians model introduced at the start, we note that a large $\sigma$ is equivalent to introducing *large noise* in directions perpendicular to the separation axis. □

We can now understand mechanistically how grokking occurs. For $\lambda$ values close enough to $\frac{1}{2}$ from below, the limiting norm $\|\boldsymbol{S}_\infty\|$ is arbitrarily large (Prop. 3.4). For large enough $\sigma$, $\boldsymbol{S}(t)$ will grow arbitrarily fast towards $\boldsymbol{S}_\infty$ (Prop. 3.5). Under these conditions, the growth rate of $b(t)$ remains bouded, and the generalization can be delayed for *arbitrarily long times*. Note that this necessitates *both* $\lambda \to \frac{1}{2}^-$ and $\sigma \to \infty$, as is also demonstrated in the right panel of Fig. 3. Interestingly, using adaptive momentum based optimizers like ADAM (Kingma & Ba, 2017), one can see significant grokking even for $\sigma = 1$, see App. B and App. F for more details.

## 4. Insights From a Simplified Model

Our main claim is that the asymptotic dynamics depend only on the separability of the training set, and that grokking occurs at the edge of linear separability. This intuition can be worked out explicitly in a much simpler setting in one dimension. In this case, separability boils down to asking whether the origin is contained between the extremal points $\min\{x_i\}$ and $\max\{x_i\}$. Therefore, all the phenomenology of the full model described in the previous sections can be captured by a training set consisting of only 2 points $x_1, x_2$, representing the points with maximal and minimal projections along $\boldsymbol{S}_\infty$. For consistency with the problem of Gaussian data, we parameterize this set as

$$x_1 = -\sigma, \qquad x_2 = \sigma(1 - 2\lambda), \qquad (11)$$

$$\mathcal{L}(S, b) = \frac{1}{2}\left(e^{Sx_1 + b} + e^{Sx_2 + b}\right),$$

so that the scale of $x_i$ is $\sigma$, and they are separable (inseparable) for $\lambda < 1/2$ ($\lambda > 1/2$). We note that the margin of the dataset from the origin has the same dependence as in the Gaussian model with $N$ points in $d$ dimensions.

The asymptotic dynamics of this model qualitatively, and sometimes quantitatively, capture the phenomenology of the full problem. The model is fully tractable analytically and the detailed analysis is presented in App. E. We summarize here the main results:

- The left panels of Fig. 3 show the geometry of the problem in $1 + 1$ dimensions. It is easily seen that the optimal SVM solution is $s = 0, b = -1$ if and only if the data is not separable, i.e. when the segment $x_2 \geq 0$

contains the origin.

- The limiting value $S_\infty$ can be easily found to be $S_\infty = \frac{1}{2(\lambda-1)} \log(1-2\lambda)$, for the separable case $\lambda < 1/2$. Indeed, it diverges logarithmically at $\lambda = 1/2$, in agreement with the numerical results of the full model presented in the upper-right panel of Fig. 1. The long time dynamics of $\|\boldsymbol{S}(t)\|$ and $b(t)$ are summarized in Table 1.

- The behavior of the loss and accuracy of the simplified model as a function of $\lambda$ is remarkably similar to that of the full model, see Fig. 8 in App. E.

**Criticality.** We note that this result bears a striking resemblance to that of Levi et al. (2023), which employed the MSE loss in a linear regression problem, again for $N$ points sampled iid from an isotropic Gaussian distribution. In their setting, the interpolation threshold is at $\lambda = 1$, in the sense that for $\lambda < 1$ the model always generalizes asymptotically, and never generalizes for $\lambda > 1$. They also found logarithmic divergence of the "grokking time" (the time difference between the times it takes for the generalization and training accuracy to reach a certain threshold). It diverges as a function of the distance from criticality as $\propto \log(1 - \sqrt{\lambda})$, which was explained in terms of a "critical slowing down" effect, arising from a vanishing eigenvalue of the data covariance near criticality. While the two problems are quite different, they both display a critical behavior near an effective interpolation threshold of the corresponding problem. We believe this is not a coincidence but rather a manifestation of a deeper relation between the behaviors of NNs in the vicinity of critical points.

## 5. Extensions of the Setup

In our original setup, we assumed $\sigma_A \ll \mu$, which allowed us to reduce the model to a $d$-dimensional problem with a bias term under constant labels. In this section, we explore what happens when this assumption does not hold. We show that while the condition $\sigma_A \ll \mu$ (or equivalently, constant labels) is essential for full generalization, grokking – in the sense of delayed generalization as well as non-monotonic test loss behavior, can still be observed provided the system remains near the critical point of linear separability.

Typically, binary classification models fully generalize only when the number of samples is sufficiently large, i.e., $N \gg d$. However, grokking is observed outside this regime, with $d/N \approx 1/2$. Here, the assumption $\sigma_A \ll \mu$ becomes particularly useful, as it ensures perfect test accuracy for any $\lambda$ below the critical point. When this assumption is relaxed, full generalization is no longer guaranteed. Nevertheless, the underlying mechanism discussed in earlier sections remains: near the critical point $\lambda = 1/2$, the model can initially reduce the loss by adjusting the direction and norm of the perpendicular weights $\boldsymbol{S}_2, ..., \boldsymbol{S}_{d+1}$, while

only modifying $\boldsymbol{S}_1$ at later times, leading to delayed generalization. In Fig. 4 we present numerical simulations supporting this behavior: While $\sigma_A = 0.01$ is small enough to closely follow the behavior of $\sigma_A = 0$, for larger values ($\sigma_A = 0.05, 0.1$), we observe that the limiting accuracy is below 1 and the loss begins to diverge at long times.

Finally, we note that while the precise dynamics depend on the data distribution and labeling scheme, similar results are expected for a broad class of linear binary classification problems, as long as $\lambda$ is near the critical point. For instance, in App. I, we analyze a generalization of the setup from another perspective, where the constant labels are explicitly modified to be discriminative. The results closely resemble those presented here, see Fig. 16.

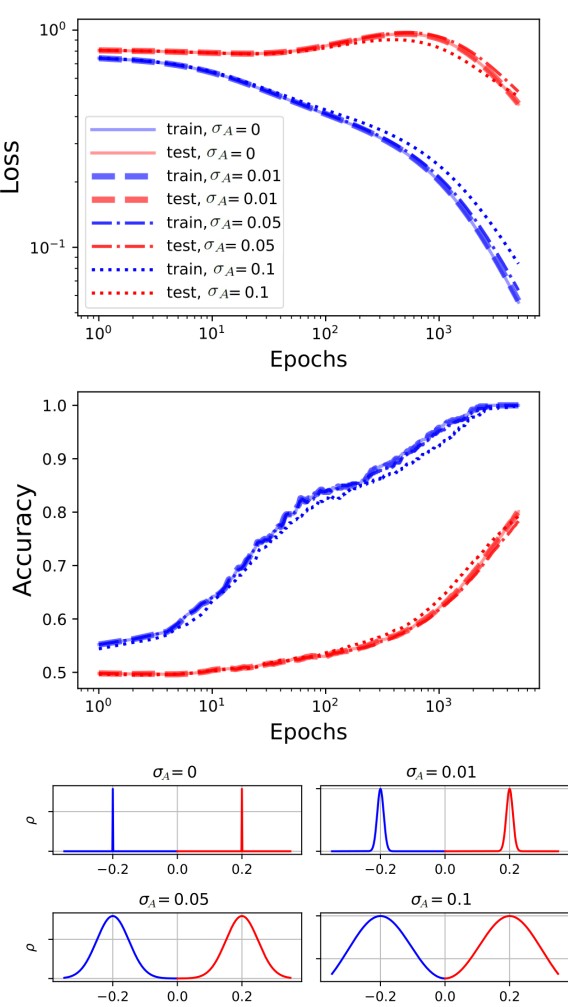

*Figure 4.* Top: The GD dynamics of the loss and accuracy for the 2-Gaussians model, with different values of $\sigma_A$. The rest of the parameters are $\lambda = 0.4$, $\mu = 0.2$, $\sigma_B = 1$, and $\eta = 0.1$. Bottom: illustration of the separation for each $\sigma_A$.

## 6. Discussion, conclusions and limitations

We studied the dynamics of gradient descent in a simple setting of logistic classification with strong noise. We have shown that in this setting, grokking occurs near a critical point in the asymptotic dynamics. Specifically, at the critical point, which occurs at $\lambda = 1/2$ in our case, the overfitting and generalizing solutions exchange stability. We showed that this non-analytic change in the asymptotic dynamics is the cause for grokking, much like in Rubin et al. (2024); Levi et al. (2023); Doshi et al. (2024), and to some extent also Humayun et al. (2024), who showed that grokking occurs near a phase transition.

Intuitively, in the vicinity of the critical point there are "flat directions" in the loss landscape. These directions may cause training to stay in the vicinity of almost-stable solutions for arbitrarily long times periods before eventually converging to the global minimum. In the physics literature, this behavior is known as "critical slowing down" (e.g. (Sethna, 2021)). In the current context, this is the mechanism of delayed generalization, which also explains the non-monotonic evolution of the generalization loss.

While we cannot show it rigorously, we conjecture that grokking is intimately related to such critical points also in different settings. In a few examples, this has been directly demonstrated, (Levi et al., 2023; Rubin et al., 2023; 2024; Humayun et al., 2024). We note that other intriguing phenomena, such as the non-monotonic dependence of asymptotic performance on model complexity, a.k.a "double descent", have also been proposed to be related to criticality, e.g. (Schaeffer et al., 2023).

If this is indeed the case, then in analogy to the theory of critical phenomena in physics, there might exist "universality classes" that have similar critical behavior, but possibly very different underlying mechanisms (Sethna, 2021). We will address this connection in future work.

**Limitations:** We considered a specific problem of linear binary classification in high dimensions. It is natural to ask how our results extend to more complex data, for instance including non-trivial correlations, hierarchical structure, or a finite sample space, as in the original observation of grokking in Power et al. (2022); Gromov (2023). While we believe the same analysis can be repeated in these instances, in the sense of (non)linear separability, we leave this to future work. In any case, we do not claim that the underlying mechanism of criticality is necessarily related to separability.

The analytic were all done in the GF limit, and while our results were verified by experiments with a finite learning rate, it may be interesting to study how large learning rates affect this setup, possibly relating to catapults (Lewkowycz et al., 2020) or the edge of stability mechanism (Cohen et al., 2022).

Lastly, we did not study the prospect of nonlinear logistic regression, which is closer to deep learning models in the wild. We believe some of our results may be generalized, provided we accept a "feature map" description of the model up to the last layer, and consider the SVM solution on the learned features.

## Acknowledgments

We thank Hillel Aharoni, Francois Charton, Amit Moscovich, Daniel Soudry and Matthieu Wyart for fruitful discussions. YBS was supported by research grant ISF 1907/22 and Google Gift grant. NL is supported by the EPFL AI4science program.

## Impact Statement

This paper presents work whose goal is to advance the field of Machine Learning. There are many potential societal consequences of our work, in particular due to the large model sizes considered in this work, but we do not feel there are specific aspects of this work with broader impacts beyond the considerations relevant to all large machine learning models.

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

# Appendix

## A. Separability and Wendel's Theorem

Wendel's theorem (Wendel, 1962) states that the probability that $N$ random vectors drawn from a distribution in $d$ dimensions are linearly separable, is

$$p = \frac{1}{2^{N-1}} \sum_{k=0}^{d-1} \binom{N-1}{k} \tag{12}$$

In relation to our work, the only assumptions required from the distribution is that

- It is symmetric around the origin, i.e. $P(\boldsymbol{x}) = P(-\boldsymbol{x})$, and

- The dataset is almost surely in general position.

We note that Eq. (12) is a the cumulative probability function of the Binomial distribution, i.e. the probability that the the number of successes is greater than $d$ out of $N-1$ attempts with success probability $\frac{1}{2}$. The central limit theorem states that in the limit of large $N, d$ the binomial distribution approaches a Gaussian, and thus the cumulative distribution function approaches the error function. Straightforward manipulations show that for large $N, d$,

$$p(\lambda) \to \frac{1}{2}\left[1 + \mathrm{erf}\left(\sqrt{d}\left(\sqrt{2\lambda} - \frac{1}{\sqrt{2\lambda}}\right)\right)\right], \qquad\qquad \lambda = \frac{d}{N} . \tag{13}$$

It is seen that for $d \to \infty$ the transition becomes infinitely sharp as a function of lambda and we have

$$\lim_{d\to\infty} p(\lambda) = \begin{cases} 0 & \lambda < \frac{1}{2} \\ \frac{1}{2} & \lambda = \frac{1}{2} \\ 1 & \lambda > \frac{1}{2} \end{cases} \tag{14}$$

See also Cover (1965) for further discussion.

## B. Relation to canonical examples

In this section, we will discuss the similarities and differences of our work with previous examples of Grokking in the literature, focusing on the seminal work of Power et. al. (Power et al., 2022). We first note that Grokking at Power et. al. is significant when the fraction of the data used for training $\alpha = N_{training}/N$ is near a critical value $\alpha_c$, in the sense that the system achieves perfect generalization if and only if $\alpha > \alpha_c$, as can be seen in Fig. 1 (center) of their paper. We expect that this non-analytic behavior in the long time limit of training will be the crucial property that underlies grokking. That is, we expect that near such points the dynamics will be slow. We note that $\alpha$ in Power et. al. is analogous to our $\lambda$ parameter, defining an effective "interpolation threshold" for the modular arithmetic problem.

Secondly, we note that a noticeable difference between our work and that of Power et. al. is that in our case, the accuracy shows a rise from the start rather than staying at chance level for a long time before generalizing. We argue that this is only a superficial discrepancy that depends on the choice of optimizer and fine-tuning of hyperparameters, and that the fundamental mechanism (that grokking occurs near critical points in which solutions exchange stability and dynamics are generically slow) is the same.

Indeed, in Fig. 5 we show that our setup is capable of grokking with accuracy staying at chance level (50%) at the start, similar to Power et al. We achieved this by using $\lambda$ values closer to half ("almost separable") and the Adam optimizer instead of vanilla gradient descent (GD). The fact that this optimizer converges faster on the training data is no coincidence: the adaptive learning rate leads to quicker convergence to large values of $|S|$ (the "memorizing solution"), maintaining accuracy at chance level until later stages, before going to large $-b$ (the "generalizing solution"). Notably, Power et al. also used Adam (or AdamW). In conclusion, although Adam can lead to a slightly "cleaner" grokking result, we explored GD because it is easier to derive analytical insights from it while, we believe, not changing the underlying mechanism of

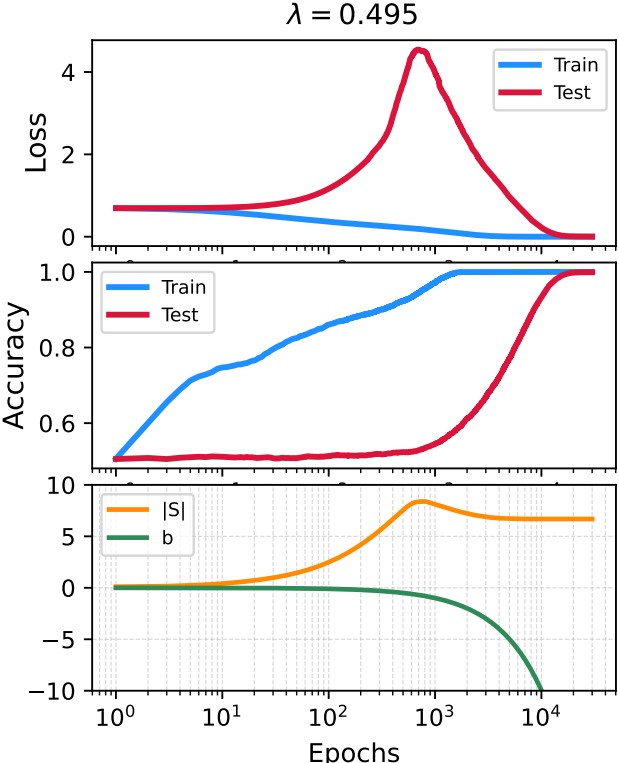

*Figure 5.* Grokking in a similar setup to the results in the main text but with ADAM optimizer (with $\beta_1 = 0.8$, $\beta_2 = 0.9$), instead of GD. The parameters are $\lambda = d/N = 0.495$, $N = 4000$ and $\sigma = 1$.

grokking. Finally, We will also note that the non-monotonicity of the test loss is also a typical sign of Grokking that can be seen in our setup (for example, compare Fig. 4 of Power et al. with the test loss in Fig. 5).

## C. Divergence of the conformal time

In the main text we have defined the "conformal time" $\tau = \int_0^t e^{b(t')} dt'$ and saw that as the result the gradient-decent trajectory of $\boldsymbol{S}(t)$ is the same as one that minimizes the exponential loss without bias: $\mathcal{L} = \frac{1}{N} \sum_{i=1}^N e^{\boldsymbol{S}^T \boldsymbol{x}_i}$. However, if $\tau$ is bounded it might reach a different fixed point. We will show now that indeed $\tau$ must diverge. First, we notice that the loss must be bounded from above: If the points are not separable (that is, there is some $\varepsilon > 0$ such that for any $\frac{\boldsymbol{S}^T}{\|\boldsymbol{S}\|}$ that we choose $\frac{\boldsymbol{S}^T}{\|\boldsymbol{S}\|} x_i > \varepsilon$ for any $i$), then it must be true since $\|\boldsymbol{S}\|$ is bounded — otherwise the loss would be infinite. If the points are separable, then $\|\boldsymbol{S}\|$ might diverge (and will, as discussed in the main text) but at some point all of the arguments of the exponent would be negative, so the loss would be trivially bounded by 1. Now, using the fact that $\frac{\partial \beta}{\partial t} = \beta \frac{\partial b}{\partial t}$ we have $\frac{\partial \beta}{\partial t} = -\eta \beta^2 \frac{1}{N} \sum_i e^{S \cdot x_i}$. Denoting $\mathcal{L}(t) < C$, we see that

$$-\frac{1}{\beta^2} \frac{\partial \beta}{\partial t} < \eta C. \tag{15}$$

We note that on the left-hand side we have a positive function (since $\frac{\partial \beta}{\partial t} < 0$). In other words, $\frac{\partial}{\partial t} \left[ \frac{1}{\beta(t)} \right] < \eta C$, so we get that

$$\frac{1}{\beta(t)} = \frac{1}{\beta(0)} + \int_0^t \frac{\partial}{\partial t} \left[ \frac{1}{\beta(t)} \right] < \frac{1}{\beta(0)} + \int_0^t \eta C = \frac{1}{\beta(0)} + \eta C t \tag{16}$$

so that $\frac{1}{\beta(t)} < 1 + \eta C t$ or, $\beta(t) > \frac{1}{1+\eta C t}$. This means that $\int_0^t \beta(t) > \int_0^t \frac{1}{1+\eta \varepsilon t}$, which diverges.

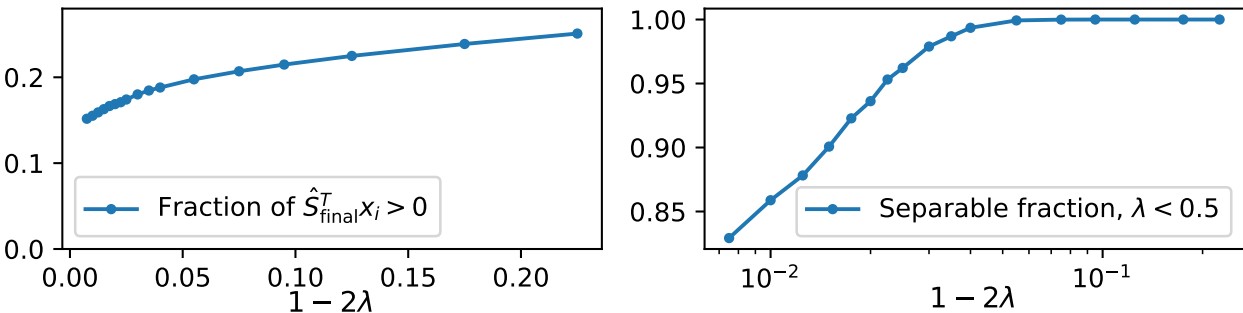

Figure 6. Left panel: The fraction of positive $\frac{\boldsymbol{S}_\infty^T}{\|\boldsymbol{S}_\infty\|}\boldsymbol{x}_i$, which goes to a constant for $\lambda = 1/2$. Right panel: The fraction of separable datasets for $\lambda < 1/2$ that were not included in the calculation.

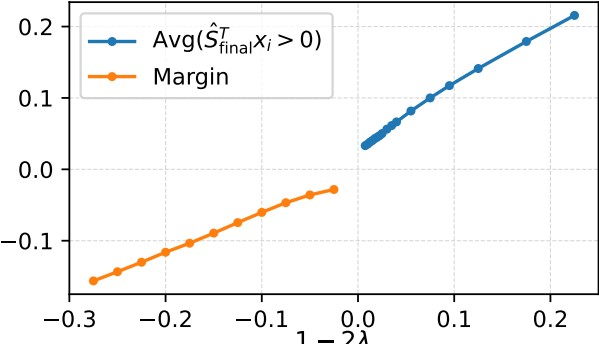

Figure 7. Numerical investigation of properties of the limiting distribution of $\boldsymbol{S}^T\boldsymbol{x}_i$, as a function of $1 - 2\lambda$ (averaged over different random configurations). In blue, we plot the average value of positive $\frac{\boldsymbol{S}_\infty^T}{\|\boldsymbol{S}_\infty\|}\boldsymbol{x}_i$, for $\lambda < 1/2$ (by minimizing $\mathcal{L} = \frac{1}{N}\sum_{i=1}^N e^{\boldsymbol{S}^T\boldsymbol{x}_i}$ ), and the margin for $\lambda > 1/2$ (using SVM).

## D. Proof that $S_\infty$ diverges for almost separable data

We look at the function

$$f(S, \{x_i\}) = \sum_{i=1}^n e^{S \cdot x_i} \qquad\qquad x, S \in \mathbb{R}^d \qquad\qquad (17)$$

We will assume $n > d$ and that the data is in general position, and that it is not separable from the origin. Since for every $S$ we must have $S \cdot x_i > 0$ for some $i$, it is easy to see that $f$ diverges when $S$ grows large in any direction. Since $f > 0$, there exists a global minimum at finite $S$.

A minimum (which is also unique under our assumptions but that's not crucial) obeys

$$\frac{\partial f}{\partial S} = \sum_{i=1}^n x_i e^{S \cdot x_i} = 0 \qquad\qquad (18)$$

If we divide this expression by $f$, we get

$$\sum_{i=1}^n p_i x_i = 0 \qquad\qquad p_i = \frac{e^{S \cdot x_i}}{f} \qquad\qquad 0 \leq p_i \leq 1, \quad \sum_i p_i = 1 \qquad\qquad (19)$$

Eq. (19) means that the origin is a convex combination of the sample points with weights $p_i$. We found that a necessary condition for the existence of a critical point at a finite $S$ is that the origin is contained in the convex hull of the sample

points. This is of course equivalent to the condition that the origin is not linearly separable from the sample data.

We want to show that if the data is almost separable, that is, if it is not separable but the origin is close to the boundary of the convex hull, then $S$ must be large. The intuition for this comes from Eq. (19): if the origin is very close to the boundary of the convex hull then some of the $p_i$'s must be very large compared to the others, which can only happen if $S$ is large.

In fact, the origin is *exactly* on the boundary of the convex hull (that is, the data is exactly on the edge of separability) if and only if for every representation of the origin as a convex combination of the sample points,

$$\sum_{i=1}^{n} q_i x_i = 0 \, , \tag{20}$$

the weights $q_i$ are non zero only for $k$ sample points, say $x_i, \ldots, x_k$, with $k \le d$, and $x_1, \ldots x_k$ are the vertices of a facet of the convex hull. This naturally leads to the definition:

**Definition**    We say that the origin is $\epsilon$-close to the boundary if there exist $k$ points $x_1, \ldots, x_k$ such that for every representation of the type of Eq. (20), the total weight assigned to $x_1, \ldots x_k$ is at least $1 - \epsilon$,

$$\sum_{i=1}^{k} q_i \ge 1 - \epsilon$$

**Theorem**    If the origin is $\epsilon$-close to the boundary of the convex hull of the sample points, then the norm of $S = \arg\min f$ is bounded from below by

$$|S| \ge \frac{1}{D} \log\left(\frac{1 - \epsilon}{\epsilon}\right)$$

where $D = \max_{ij} |x_i - x_j|$ is the diameter of the data.

*Proof.*    We divide the points to two groups: $A = \{x_1, \ldots x_k\}$, and $B = \{x_{k+1}, \ldots x_N\}$. Since the origin is $\epsilon$-close, the ratio of the weights of the two groups is bounded by

$$\frac{\sum_{i \in A} p_i}{\sum_{i \in B} p_i} \ge \frac{1 - \epsilon}{\epsilon} \tag{21}$$

Consider now the convex combination Eq. (19). Using Jensen's inequality, we can bound the relative weights of the second group by

$$\sum_{i \in B} e^{S \cdot x_i} \ge (N - k) e^{S \cdot \bar{x}} \, , \qquad \text{with} \qquad \bar{x} = \frac{1}{N - k} \sum_{i \in B} x_i \tag{22}$$

where $\bar{x}$ is the average of the points in the second group. Therefore, the ratio is bounded by

$$\frac{\sum_{i \in A} p_i}{\sum_{i \in B} p_i} = \frac{\sum_{i \in A} e^{S \cdot x_i}}{\sum_{i \in B} e^{S \cdot x_i}} \le \frac{\sum_{i \in A} e^{S \cdot x_i}}{(N - K) e^{S \cdot \bar{x}}} \le \frac{k}{N - k} e^{|S|D} \le \frac{k}{N - k} e^{|S|D} \tag{23}$$

Combining Eq. (21) and Eq. (23) we get

$$\frac{1 - \epsilon}{\epsilon} \le \frac{k}{N - k} e^{|S|D} \qquad \Rightarrow \qquad |S| \ge \frac{1}{D} \log\left(\frac{1 - \epsilon}{\epsilon} \cdot \frac{N - k}{k}\right) \tag{24}$$

Since $k \le d$, we also have $N - k \ge n - d$.

Note that the same convexity argument would work also for logisitic loss $f = \sum_i \ell(S \cdot x_i)$, $\ell(z) = \log(1 + e^z)$, or any other monotonic and convex $\ell$. In this case the only difference is that the log function should be replaced the inverse of $\ell$. $\qquad \square$

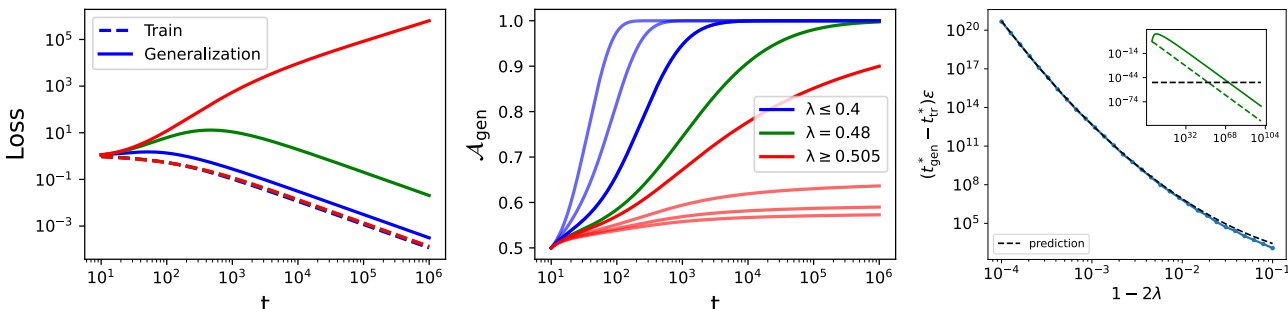

*Figure 8.* **Simplified model** (left, center) Loss and accuracy for different $\lambda$ and $\sigma = 5$. dotted/solid lines represent the training/generalization respectively. (right) Grokking time (the time difference between the time it takes for the training and generalization loss to reach a certain threshold, $\varepsilon = 10^{-50}$ in this case) for $\sigma = 20$. The data is in very good agreement with the prediction. (inset) how grokking time is calculated for $\varepsilon = 10^{-50}$ and $\lambda = 10^{-4}$.

## E. Details of the Simplified Model

### E.1. Justification and Relation to the Full Model

We will provide here supplemental results regarding the justification of the simplified model (by numerical comparison to the full model). To obtain the results we average over different random realizations: Assuming the so-called "self averaging" property, we know that the average over a large number of finite systems should give us the same result as the infinite system (where $N, d \to \infty$ and the ratio is constant).

In the non-separable case, $\boldsymbol{S}_\infty$ can be found by any optimizer that minimizes the loss $\mathcal{L} = \sum e^{\boldsymbol{S}^T \boldsymbol{x}_i}$. We note that when getting close to the transition point, for any finite-sized system we have some probability of getting a separable set (even though $\lambda < 1/2$), see Eq. (12). In this case, we just ignore the the result: In the right panel of Fig. 6 we present the fraction of realizations that are separable. This will probably introduce some bias into the results which is likely the cause of the fact that the average of positive samples (and similarly, the margin) does not go exactly to zero for $\lambda \to 1/2$ (see Fig. 7). In the left panel of Fig. 6 we present the fraction of positive $\frac{\boldsymbol{S}_\infty^T}{\|\boldsymbol{S}_\infty\|} \boldsymbol{x}_i$ . Interestingly, it does not go to zero but to some positive constant, implying that there is a singularity in the density of $\frac{\boldsymbol{S}_\infty^T}{\|\boldsymbol{S}_\infty\|} \boldsymbol{x}_i$ at $\lambda = 1/2$.

### E.2. Analytical Predictions

Here, we provide the full analysis of the model presented in Sec. 4, for a single point fixed at $x_1 = -1$, and a second point $x_2 = x = 1 - 2\lambda$, where $\lambda = d/N$.

The gradient flow equations in conformal time are given by

$$\frac{\partial S}{\partial \tau} = -\frac{\eta}{2} \left( x e^{Sx} - e^{-S} \right) = -\frac{\eta}{2} \left( (1-2\lambda)e^{S(1-2\lambda)} - e^{-S} \right), \tag{25}$$
$$\frac{\partial b}{\partial \tau} = -\frac{\eta}{2} \left( e^{Sx} + e^{-S} \right) = -\frac{\eta}{2} \left( e^{S(1-2\lambda)} + e^{-S} \right).$$

While there exist analytical solutions for Eq. (25), they do not necessarily provide any intuition, and so we find it better to begin by investigating three special representative cases:

1. $x_2 = 1$ (non-separable).

2. $x_2 = 0$ (marginally non-separable).

3. $x_2 = -1$ (separable).

For $x = 1$ (A), the data is entirely non-separable in one dimension and the conformal time solutions are

$$S(\tau) = \log\left(\tanh\left(\frac{\eta\tau}{2} + \tanh^{-1}\left(e^{S_0}\right)\right)\right), \tag{26}$$

$$b(\tau) = b_0 + \log\left(\frac{\tanh\left(2\tanh^{-1}\left(e^{S_0}\right)\right)\cosh\left(2\tanh^{-1}\left(e^{S_0}\right)\right)}{\tanh\left(\eta\tau + 2\tanh^{-1}\left(e^{S_0}\right)\right)\cosh\left(\eta\tau + 2\tanh^{-1}\left(e^{S_0}\right)\right)}\right),$$

in which case the generalization accuracy reaches 1 for $\tau \to \infty$, as $b(\tau)$ grows faster than $S(\tau)$ with conformal time.

For $x_2 = 0$ (B), the equations in conformal time become:

$$\frac{\partial S}{\partial \tau} = \frac{\eta}{2}e^{-S}, \qquad \frac{\partial b}{\partial \tau} = -\frac{\eta}{2}(e^{-S} + 1) \tag{27}$$

By solving for $S$ and plugging into $\frac{\partial b}{\partial \tau}$, we immediately get

$$S = \log\left(e^{S_0} + \frac{\eta}{2}\tau\right), \qquad b = -\log\left(e^{S_0} + \frac{\eta}{2}\tau\right) - \frac{\eta}{2}\tau + S_0 + b_0. \tag{28}$$

Using the fact that $e^b = \frac{\partial \tau}{\partial t}$, we get that $\frac{\partial \tau}{\partial t} = \frac{e^{-\frac{\eta}{2}\tau}}{e^{S_0} + \frac{\eta}{2}\tau}e^{S_0 + b_0}$, and taking another integral, we get that $e^{\frac{\eta}{2}\tau}\left[e^{S_0} - 1 + \frac{\eta}{2}\tau\right] = e^{S_0 + b_0}\frac{\eta}{2}t + (e^{S_0} - 1)$. Taking the inverse of this, we finally get

$$\tau = \frac{2}{\eta}\left[W_0\left(\left(e^{S_0 + b_0}\frac{\eta}{2}t + (e^{S_0} - 1)\right)e^{e^{S_0} - 1}\right) - e^{S_0} + 1\right], \tag{29}$$

where $W_0$ is the Lambert W function. We note that for large $t$ we have $\tau \sim \log(t)$, and therefore $b \sim -\log(t)$, $S \sim \log(\log(t))$, so it is interesting to note that in the critical point we still have $\lim_{t\to\infty} S(t)/b(t) = 0$ (i.e., accuracy goes to 1), even though S diverges.

Finally, for $x = -1$ (C), the data is fully separable in one dimension and the solution in conformal time is given by

$$S(\tau) = S_0 + \log\left(1 + \eta\tau e^{-S_0}\right), \qquad b(\tau) = b_0 - \log\left(1 + \eta\tau e^{-S_0}\right), \tag{30}$$

showing that the accuracy is bounded at $\mathcal{A}_{\text{gen}}^\infty = \frac{1}{2}\left(1 + \text{erf}\left(\frac{1}{\sqrt{2}}\right)\right)$ agreeing with Item 1 for $M = 1$.

For completeness, we report here the full solution, as a function of the conformal time $\tau = \int_0^t e^{b(t)}dt$ of Eq. (25). We define

$$f(y) = -\frac{xe^{y(x+2)}\,{}_2F_1\left(1, 1 + \frac{1}{x+1}; 2 + \frac{1}{x+1}; e^{(x+1)y}x\right)}{x+2} - e^y, \tag{31}$$

then the solution for $S(\tau)$ is given by the inverse function $f^{-1}(u)$ evaluated at

$$u = -\frac{xe^{S_0(x+2)}\,{}_2F_1\left(1, 1 + \frac{1}{x+1}; 2 + \frac{1}{x+1}; e^{S_0(x+1)}x\right)}{x+2} - \frac{\tau}{2} - e^{S_0}, \tag{32}$$

as

$$S(\tau) = f^{-1}\left(-\frac{xe^{S_0(x+2)}\,{}_2F_1\left(1, 1 + \frac{1}{x+1}; 2 + \frac{1}{x+1}; e^{S_0(x+1)}x\right)}{x+2} - \frac{\eta\tau}{2} - e^{S_0}\right). \tag{33}$$

The solution for $b(\tau)$ is obtained simply by integrating Eq. (25), resulting in

$$b(\tau) = \frac{1}{x}\left[b_0 x - \log\left(1 - e^{(1+x)f^{-1}\left(-e^{S_0} - \frac{e^{S_0(2+x)}x\,_2F_1\left(1,1+\frac{1}{x+1};2+\frac{1}{x+1};e^{S_0(1+x)}x\right)}{2+x}\right)x}\right)\right.$$

$$+ \log\left(1 - e^{(1+x)f^{-1}\left(-e^{S_0}-\frac{\eta\tau}{2}-\frac{e^{S_0(2+x)}x\,_2F_1\left(1,1+\frac{1}{x+1};2+\frac{1}{x+1};e^{S_0(1+x)}x\right)}{2+x}\right)x}\right)$$

$$+ xf^{-1}\left(-e^{S_0}-\frac{e^{S_0(2+x)}x\,_2F_1\left(1,1+\frac{1}{x+1};2+\frac{1}{x+1};e^{S_0(1+x)}x\right)}{2+x}\right)$$

$$\left.- xf^{-1}\left(-e^{S_0}-\frac{\eta\tau}{2}-\frac{e^{S_0(2+x)}x\,_2F_1\left(1,1+\frac{1}{x+1};2+\frac{1}{x+1};e^{S_0(1+x)}x\right)}{2+x}\right)\right]. \tag{34}$$

While these solutions may not necessarily be instructive in this form, appropriate limits can be taken in order to obtain the results in the main text.

### E.3. Grokking time in the simplified model

We can define $t_{\text{tr}}^*$, $t_{\text{gen}}^*$ as the times it would take for the training and generalization loss to reach some threshold $\varepsilon$. We can find $t_{\text{tr}}^*$ by solving $\mathcal{L}_{\text{tr}} = \frac{1}{2}e^b\left(e^{-S} + e^{S(1-2\lambda)}\right) = \varepsilon$. We will assume that $\sigma$ is large enough such that $S = S_\infty$ from the start (as discussed in the main text, $\sigma$ increase the rate that S goes to its final value). Therefore, we plug $S_\infty = -\log(1-2\lambda)$, and find that for $\lambda$ which is close enough to $1/2$, the loss is approximately given by $\mathcal{L}_{\text{tr}} = \frac{1}{2}e^b$. Comparing to $\varepsilon$ and plugging the long-time limit $b = -\log(\frac{\eta}{2}t)$, we find that

$$t_{\text{tr}}^* = \frac{1}{\eta\varepsilon}. \tag{35}$$

Similarly, using the generalization loss $\mathcal{L}_{\text{gen}} = e^b e^{S^2/2}$ we can find that

$$t_{\text{gen}}^* = \frac{2}{\eta\varepsilon}e^{\frac{1}{2}\log^2(1-2\lambda)} \tag{36}$$

It is already clear that for any finite $\varepsilon$, $t_{\text{gen}}^* - t_{\text{tr}}^*$ diverges. We can also obtain an $\varepsilon$-independent property by noting that

$$\sqrt{\log(t_{\text{gen}}^*/t_{\text{tr}}^*)} = \frac{1}{\sqrt{2}}\log(1-2\lambda), \tag{37}$$

which is verified numerically in Fig. 9.

It is interesting to note that in the conformal time, we have $b \approx -\tau$, and therefore can repeat this calculation and obtain

$$\tau_{\text{tr}}^* = -\log(\eta\varepsilon), \ \tau_{\text{gen}}^* = \frac{1}{2}\log^2(1-2\lambda) - \log\frac{\eta}{2}\varepsilon. \tag{38}$$

In this case, the result is a bit more natural since now the time difference (instead of ratio) becomes $\varepsilon$-independent:

$$\tau_{\text{gen}}^* - \tau_{\text{tr}}^* \approx \frac{1}{2}\log^2(1-2\lambda). \tag{39}$$

We note that this result still depends on $\varepsilon$ implicitly, in the sense that our assumption that $S = S_\infty$ is true only for long-times, or $\varepsilon$ which is small enough.

### E.4. Calculation of the subleading term in the separable case

We now consider $x_2 < 0$ but close to zero (that is, we are in a separable case where $M = -x_2$ is the margin). We know that $w$ diverges at long times as $w \approx \frac{M}{1+M^2}\log\left[\frac{\eta}{2}(1+M^2)t + 1\right]$. We will now denote

$$u \equiv w - \frac{M}{1+M^2}\log\left[\frac{\eta}{2}(1+M^2)t + 1\right]$$

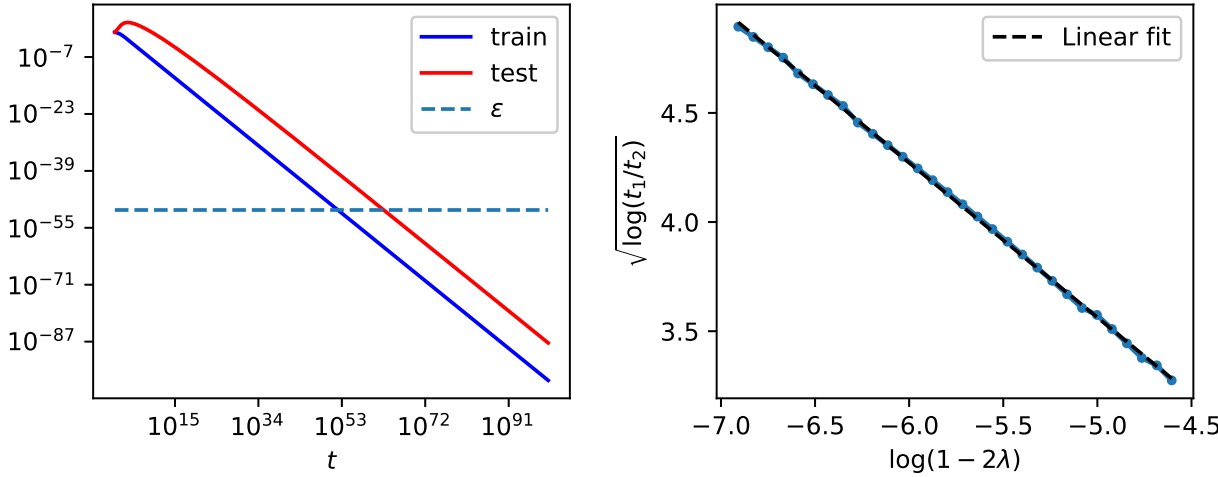

*Figure 9.* Numerical evidence for the Grokking time in the simplified model. In the left panel, we demonstrate for $1 - 2\lambda = 0.001$ how the Grokking time is calculated: $t_{\text{tr}}^*$, $t_{\text{gen}}^*$ are calculated by finding the intersection of the loss with some $\varepsilon$. In the right panel we plot $\sqrt{\log(t_{\text{gen}}^*/t_{\text{tr}}^*)}$ versus $\log(1 - 2\lambda)$ numerically, and show that the result is linear with slope $\approx \frac{1}{\sqrt{2}}$, in agreement with the prediction of Eq. (37)

as the difference from the diverging term. The equation for $u$ is therefore

$$\frac{\partial u}{\partial t} = -\frac{\eta}{2} e^b \left( -e^{x_1 \left( u + \frac{M}{1+M^2} \log\left[\frac{\eta}{2}(1+M^2)t\right]\right)} - M e^{-M\left(u + \frac{M}{1+M^2} \log\left[\frac{\eta}{2}(1+M^2)t\right]\right)} \right) - \frac{M}{1+M^2} \frac{1}{t}.$$

Plugging the (long-time) solution for the bias, $b \approx -\frac{1}{M^2+1} \log\left[\frac{\eta}{2}(1+M^2)t + e^{-(1+M^2)b_0}\right]$ (where $b_0 = 0$ in our case), we get

$$\frac{\partial u}{\partial t} = -x_1 \frac{\eta}{2} e^{x_1 u} \left(\frac{\eta}{2}(1+M^2)t + 1\right)^{\frac{x_1 M - 1}{1+M^2}} + \left(e^{-Mu} - 1\right) \frac{M}{(1+M^2)t + \frac{2}{\eta}}.$$

For $M \approx 0$, we note that the second term is $O(M^2)$, and by neglecting it we get

$$u = -\frac{1}{x_1} \log \left( \frac{x_1^2}{x_1 M + M^2} \left(\frac{\eta}{2}(1+M^2)t + 1\right)^{\frac{x_1 M + M^2}{1+M^2}} - \frac{x_1^2}{x_1 M + M^2} + 1 \right).$$

For $t \to \infty$, and neglecting the other $O(M^2)$ terms, we finally get

$$u \approx -\frac{1}{x_1} \log \left(\frac{x_1}{x_2}\right).$$

Remarkbly, this is identical to the result of the in the $x_2 > 0$ case.

## F. Impact of different parameters

Here we present supplemental results for Sections Sec. 4.

### F.1. The Variance Scale $\sigma$

Here we provide additional information regarding the effect of $\sigma$ different than 1. In particular, we will show that increasing $\sigma$ can make grokking more apparent (but only up to a certain point). We will first assume that $\sigma = 1$ at the start, and investigate how taking $\tilde{x}_i = \sigma x_i$ changes the dynamics in comparison to that case. We will begin with the non-separable

case ($\lambda < 1/2$). Recalling that the equations for gradient flow in our model are given by Eq. (7), this results in

$$\frac{\partial \boldsymbol{S}}{\partial t} = -\sigma \frac{\eta}{N} e^b \sum_i e^{\boldsymbol{S}^T \sigma \boldsymbol{x}_i} \boldsymbol{x}_i, \qquad \frac{\partial b}{\partial t} = -\frac{\eta}{N} e^b \sum_i e^{\boldsymbol{S}^T \sigma \boldsymbol{x}_i}. \tag{40}$$

We can now absorb $\sigma$ into $\boldsymbol{S}$ by denoting $\tilde{\boldsymbol{S}} \equiv \sigma \boldsymbol{S}$ and investigate how it affects the dynamics of $\tilde{\boldsymbol{S}}$, and the generalization loss and accuracy as a function of $\tilde{\boldsymbol{S}}$. First, the GD equations become

$$\frac{\partial \tilde{\boldsymbol{S}}}{\partial t} = -\sigma^2 \frac{\eta}{N} e^b \sum_i e^{\tilde{\boldsymbol{S}}^T \boldsymbol{x}_i} \boldsymbol{x}_i, \qquad \frac{\partial b}{\partial t} = -\frac{\eta}{N} e^b \sum_i e^{\tilde{\boldsymbol{S}}^T \boldsymbol{x}_i}. \tag{41}$$

We note that the generalization loss and accuracy in Eqs. (3) and (4) are the same except they are now a function of $\left\|\tilde{\boldsymbol{S}}\right\|$ instead of $\|\boldsymbol{S}\|$ (being a function of $\sigma \|\boldsymbol{S}\|$). Since the equation for $\frac{\partial \tilde{\boldsymbol{S}}}{\partial t}$ is just multiplied by a factor $\sigma^2$, the limiting value of $\tilde{\boldsymbol{S}}_\infty$ would be the same as for the $\sigma = 1$ case, but it will reach it at a *faster rate*. To sum up, obtaining the dynamics of the loss and accuracy when $\sigma$ is larger than one can be done by using the same Eqs. (3) and (4), but also (A) Increasing the starting condition of $\boldsymbol{S}_0$ by a factor of $\sigma$, and (B) Multiply only the learning rate of the spatial part by a factor of $\sigma^2$. If $\sigma$ is large enough, we can go to the fixed point of $\|\boldsymbol{S}\|$ as fast as we want, enabling the appearance of Grokking (if also the limiting value of $\|\boldsymbol{S}\|$ is large, which happens when we are on the edge of being separable).

Finally, we will also investigate the effect of $\sigma$ in the separable regime ($\lambda > 1/2$). Now we can use **??** and Item 2, where we only need to consider how $\sigma$ changes the margin $M$. Since it is obtained from the equation $\frac{\boldsymbol{S}^T}{\|\boldsymbol{S}\|} x_m = -M$, we can see that the new margin will be larger by $\sigma$ than the old one, i.e., $\tilde{M} = \sigma M$. Plugging this in the expression for the accuracy in Prop. 3.2, we get that the accuracy is now

$$\lim_{t \to \infty} \mathcal{A}_{\text{gen}} \approx \frac{1}{2} \left[ 1 + \text{erf} \left( \frac{1}{\sigma^2 M \sqrt{2}} \right) \right]. \tag{42}$$

where we note that the argument inside the erf is smaller in a factor of $\sigma^2$, drastically reducing the limiting accuracy.

### F.2. Optimizer

The effect of changing the optimizer to Adam is demonstrated in Fig. 11. We note that the fact that adaptive-type optimizers change each learning rate individually based on past gradients, leads the dynamics faster in the direction of $\|\boldsymbol{S}\|$, relatively to $b$. The fact that it makes $\|\boldsymbol{S}\|$ change faster (and not slower) than $b$, is probably related to the fact that $\boldsymbol{S}$ is a vector in high dimension: Moving each component of such vector will result in a change of the norm in a rate that is proportional to $\sqrt{d}$, but this may need further investigation. We will also note that using a different optimizer for the non-separable region where, will lead to a different solution than the hard margin SVM, as is also discussed by Soudry et al. (Soudry et al., 2018). This means that the results we developed in the main text will not hold, but we can still expect to obtain accuracy smaller than one since $\|\boldsymbol{S}\|$ diverges, as indeed can be seen in Fig. 11.

### F.3. Initial conditions

As discussed in the main text, changing the initial conditions can change the monotonicity of the generalization loss and accuracy: See Fig. 12, Fig. 13 below.

### F.4. Different loss

In the main text, we showed that we can use the exponential instead of the CE loss, since it will converge to it at late times. Here we provide numerical evidence that indeed Grokking could be seen, even when taking from the beginning just the exponential loss $\mathcal{L} = \frac{1}{N} \sum_i e^{\boldsymbol{S}^T x_i + b}$: See Fig. 14.

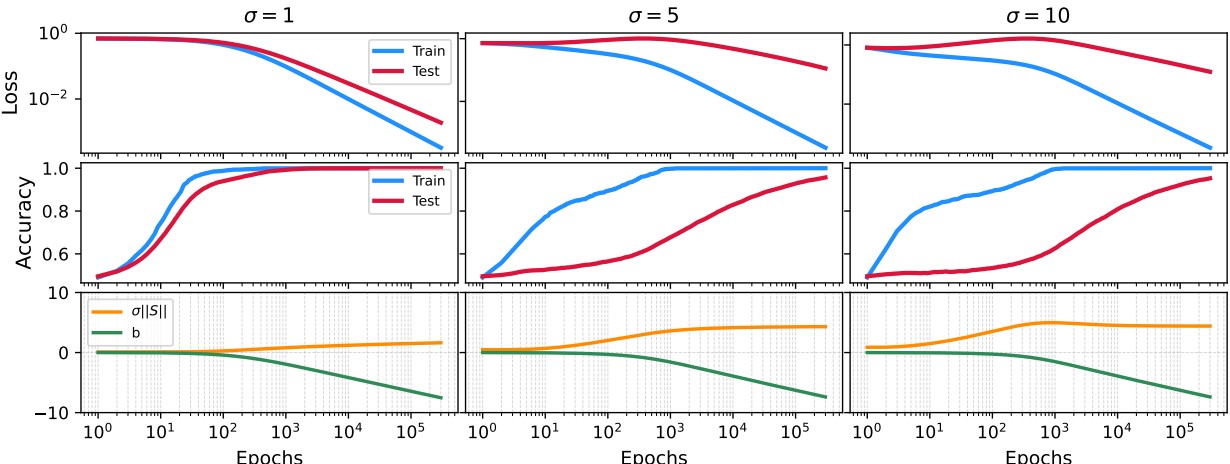

*Figure 10.* Gradient descent dynamics for three different values of $\sigma$, all for $\lambda = 0.48$. The top panels show the loss and accuracy for the train and test datasets, while the bottom panels present $b$ and the norm of $\boldsymbol{S}$. Except for $\sigma$, the parameters are the same as in Fig. 1. We can see that increasing $\sigma$ makes the grokking more apparent at start, but then saturates (that is, increasing $\sigma$ will not increase the "grokking time" anymore).

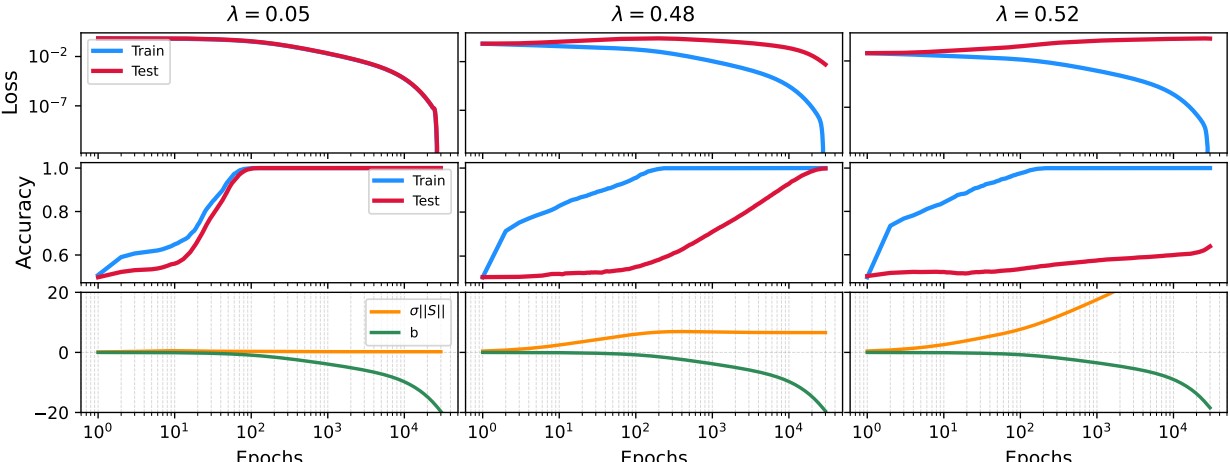

*Figure 11.* Dynamics, using ADAM optimizer with PyTorch's default parameters. The setup is the same as Fig. 1, except for the fact that $\sigma = 1$ now instead of 5. Significant Grokking can be seen even though the value of $\sigma$ is not large.

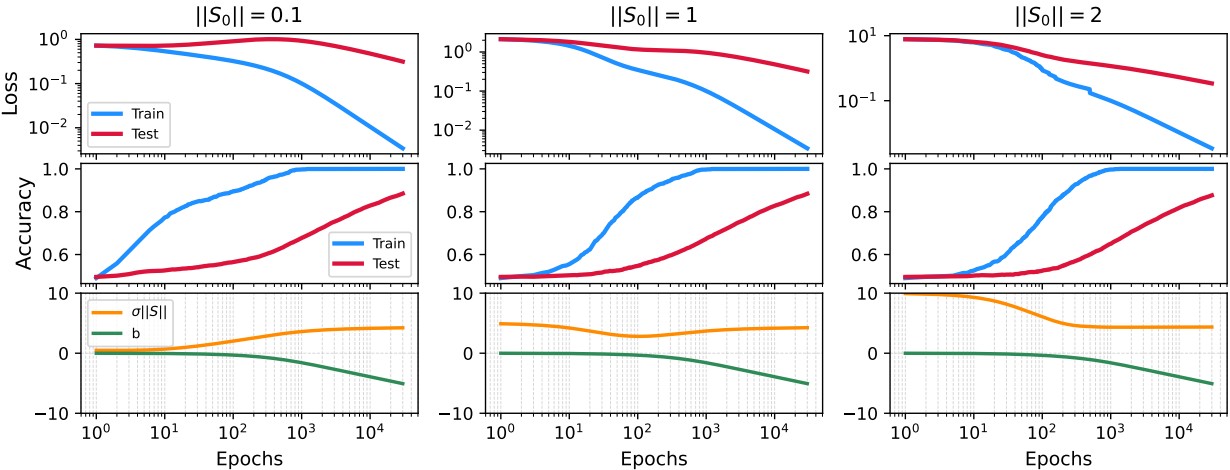

Figure 12. Gradient descent dynamics for $\lambda = 0.48$ and for three different values of starting norm, $\|S_0\|$. Except for this, the setup is the same as Fig. 1. We can see that the non-monotonicity of the loss can be affected by the starting condition.

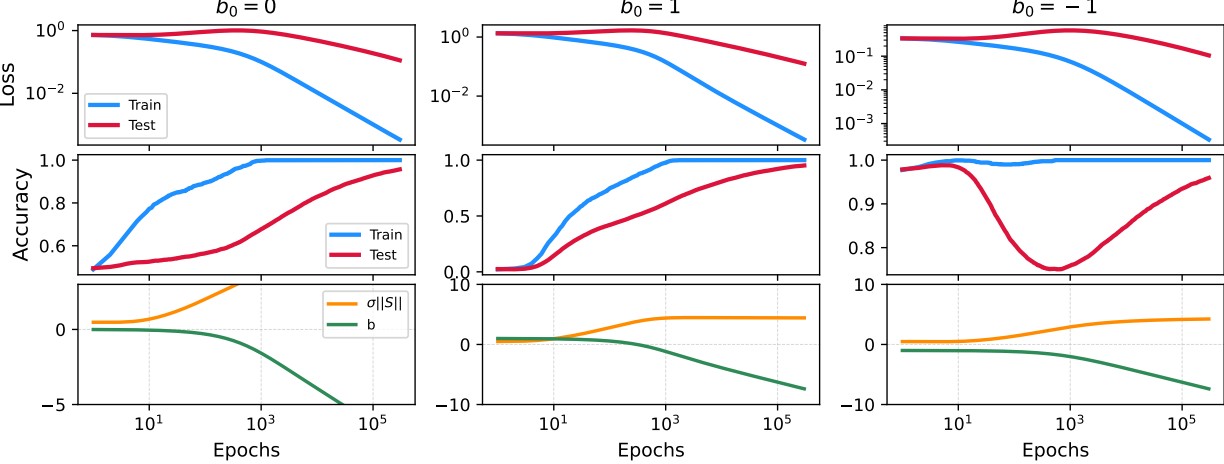

Figure 13. Gradient descent dynamics for $\lambda = 0.48$ and for three different values of $b$. Except for this, the setup is the same as Fig. 1.

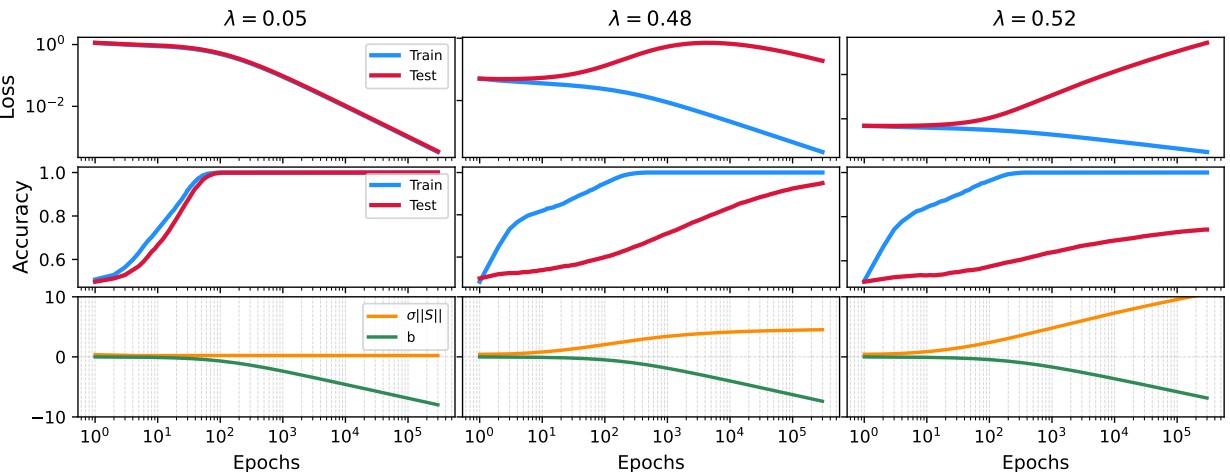

*Figure 14.* Gradient descent dynamics for a setup which is the same as Fig. 1, but with the exponent loss $\mathcal{L} = \frac{1}{N} \sum_i e^{\boldsymbol{S}^T x_i + b}$. That is, the loss is strictly the exponent loss at any time (and not just converge to the exponent loss at long times, as the CE loss). Clearly, we can see that the behavior of the Grokking is similar.

## G. Different input data distributions

As discussed in the main text, our results hold for any data distribution that is symmetric around the origin. Since the underlying mechanism only requires that the data is on the verge of separability (in which case $|\boldsymbol{S}_\infty|$ diverges). As we discuss in App. A, in the limit $d, N \to \infty$, $\lambda = 1/2$ is the critical value below which the dataset is almost surely inseparable. Therefore, the analysis and resulting behavior, including the occurrence of Grokking and the critical point of $\lambda$ should hold for any symmetric distribution.

To demonstrate this, we compare three input distributions at $\lambda = 0.45$ in Fig. 15: (1) The isotropic Gaussian input (as discussed in the main text), (2) Non-isotropic Gaussian inputs, generated using a covariance matrix with eigenvalues that follows the scaling law $\lambda_n = \frac{\lambda_0}{n^\alpha}$, with $\alpha = 1.5$. (3) Mixture of Gaussians $\mathcal{N}(\mu = \pm 1, \sigma = 0.25)$. We notice that $\sigma$ in this context (and its effect on Grokking described in Sec. 4) could also be easily generalized for any distribution, by simply multiplying all of the inputs by a factor of $\sigma$.

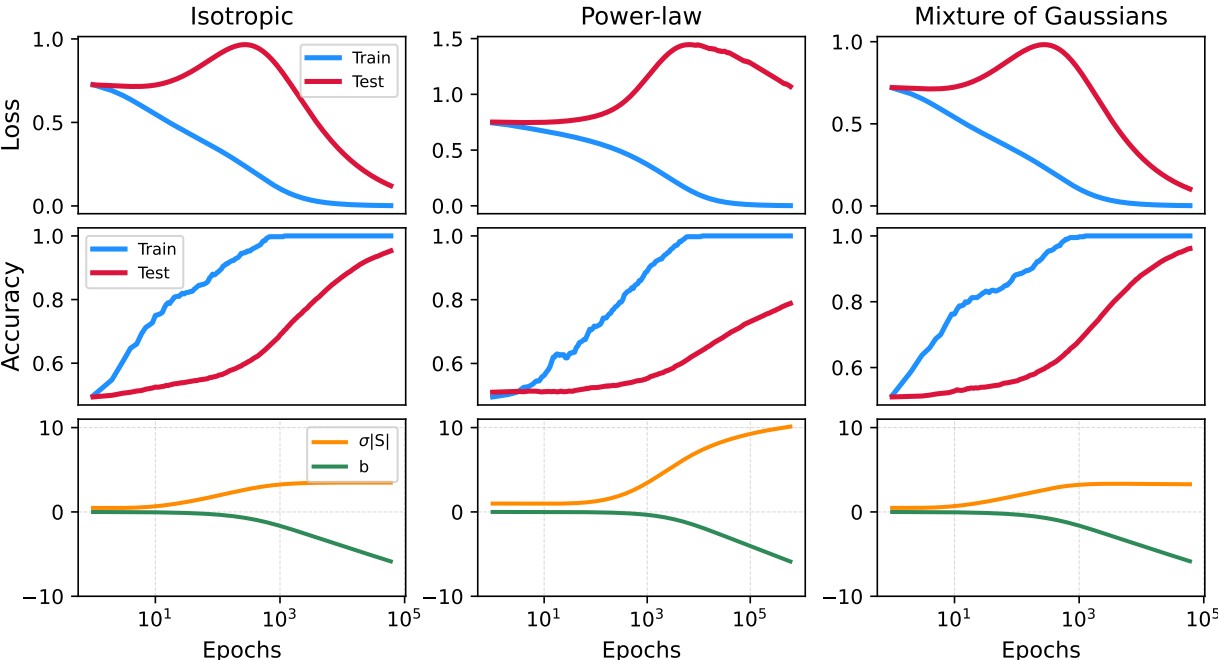

*Figure 15.* Grokking for three different input data distributions: Isotropic Gaussian (left), Gaussian with covariance whose eigenvalues follow a power-law scaling (middle), and uniform distribution (right). The parameters are: $d = 180$, $N = 400$ ($\lambda = 0.45$), and the optimizer is gradient-decent. Left panel: Isotropic Gaussian with $\sigma = 5$, as appears in the main text. Middle panel: Gaussian with eigenvalues that follow $\lambda_n = \lambda_0/n^\alpha$, where $\alpha = 1.5$. The normalization factor $\lambda_0$ is chosen such that $\sum_n \lambda_n = \sigma \cdot d$, where here $\sigma = 10$. Right panel: each element in the input vector is chosen from a mixture of Gaussian distribution, with $\mu_{1,2} = \pm 1$ and $\sigma_{1,2} = 0.25$. After sampling, the input was multiplied by 5, as a generalization of the original $\sigma$.

## H. Reduction to a d-dimensional model with bias.

In this section we will prove the equivalence between the d+1-dimensional two-Gaussians model and the d-dimensional model with bias. Consider the $d + 1$-dimensional classification problem with no bias:

$$\boldsymbol{x}_i = (y_i \mu, x_{i,2}, x_{i,3}, ..., x_{i,d+1}), \tag{43}$$

where $y_i = \pm 1$ are the labels, $\mu$ is some constant and $x_{i,j} \sim \mathcal{N}(0, \sigma_B)$. The loss is $\mathcal{L} = \frac{1}{N} \sum_i \ell_i$ where $\ell_i = \ell(y_i \boldsymbol{S} \cdot \boldsymbol{x}_i)$ is some loss. Taking $\boldsymbol{x}_i \to y_i \boldsymbol{x}_i$, we get

$$\boldsymbol{x}_i = (\mu, y_i x_{i,2}, y_i x_{i,3}, ...), \tag{44}$$

but the distribution of $y_i x_{i,j}$ stays the same since it is symmetrical, so we can assume WLOG that $y_i = 1$. The dynamics is

$$\frac{\partial \boldsymbol{S}}{\partial t} = -\eta \frac{1}{N} \sum_i \frac{\partial \ell(\boldsymbol{S} \cdot \boldsymbol{x}_i)}{\partial(\boldsymbol{S} \cdot \boldsymbol{x}_i)} \boldsymbol{x}_i. \tag{45}$$

We now denote $\tilde{\boldsymbol{x}}_i = \frac{1}{\mu} \boldsymbol{x}_i$, $\tilde{\boldsymbol{S}} = \mu S$ (so that $\boldsymbol{S} \cdot \boldsymbol{x}_i = \tilde{\boldsymbol{S}} \cdot \tilde{\boldsymbol{x}}_i$), and get

$$\frac{\partial \tilde{\boldsymbol{S}}}{\partial t} = -\eta \mu^2 \frac{1}{N} \sum_i \frac{\partial \ell(\boldsymbol{S} \cdot \boldsymbol{x}_i)}{\partial(\boldsymbol{S} \cdot \boldsymbol{x}_i)} \tilde{\boldsymbol{x}}_i, \tag{46}$$

which is the same except for a $\mu^2$ learning rate factor. Since $\tilde{\boldsymbol{x}}_i = (1, \frac{1}{\mu} x_{i,2}, \frac{1}{\mu} x_{i,3}, ...)$, we can define also the scalar $b$ and the $d$ dimensional vectors $\overline{\boldsymbol{S}}, \overline{\boldsymbol{x}}$, by

$$\tilde{\boldsymbol{S}} \equiv (b, \overline{\boldsymbol{S}}_1, \overline{\boldsymbol{S}}_2, ..., \overline{\boldsymbol{S}}_d), \ \tilde{\boldsymbol{x}}_i \equiv (1, \overline{\boldsymbol{x}}_{i,1}, ..., \overline{\boldsymbol{x}}_{i,d}) \tag{47}$$

We can see that

$$\frac{\partial b}{\partial t} = -\eta \mu^2 \frac{1}{N} \sum_i \frac{\partial \ell(\overline{\boldsymbol{S}} \cdot \overline{\boldsymbol{x}}_i + b)}{\partial(\overline{\boldsymbol{S}} \cdot \overline{\boldsymbol{x}}_i + b)}, \tag{48}$$

$$\frac{\partial \overline{\boldsymbol{S}}}{\partial t} = -\eta \mu^2 \frac{1}{N} \sum_i \frac{\partial \ell(\overline{\boldsymbol{S}} \cdot \overline{\boldsymbol{x}}_i + b)}{\partial(\overline{\boldsymbol{S}} \cdot \overline{\boldsymbol{x}}_i + b)} \overline{\boldsymbol{x}}_i \tag{49}$$

Which is exactly the dynamics of the $d$ model with bias, where all of the labels are the same. We also note that $\sigma = \sigma_B / \mu$, where $\sigma$ is factor of the std of $\overline{\boldsymbol{x}}_i$.

## I. Extension of the model from another perspective

In this section, we begin with the constant label model and extend it explicitly to a model where only a fraction $r$ of the samples are the same (so that for $r = 1$ we get the same model studied throughout the paper). We show that grokking can still be observed. Specifically, we classify a point $\boldsymbol{x}_i$ as label -1 if its first coordinate $x_{i,1}$ exceeds a threshold $\mu$, and as label 1 otherwise. Here, $\mu = Q(r)$, where $Q$ is the Gaussian quantile function (inverse CDF) and $r \in [0, 1]$ is a fraction. We note that for $r < 1$, only a fraction $r$ of points are labeled -1, while the remaining $1 - r$ are labeled 1.

The same reasoning as the previous sections can be applied here: near the critical point $\lambda = 1/2$, the model could reduce the loss first by changing the direction of $\boldsymbol{S}$, and only then modifying $b$, leading to delayed generalization. However, unlike the constant-labeling case, full generalization is not possible even for $\lambda > 1/2$, since the optimal solution is not obtained by taking the separating hyperplane to infinity. We can see this by studying the expression for the generalization accuracy (see App. I.1) given by

$$\mathcal{A}_{\text{gen}} = \frac{1}{2} \left[ 1 + \frac{1}{\sqrt{2\pi\sigma^2}} \int_{-\infty}^{\infty} dx_1 e^{-\frac{1}{2\sigma^2} x_1^2} \text{sign}(x_1 - \mu) \right.$$

$$\left. \times \text{erf} \left( \frac{1}{\sqrt{2}} \frac{x_1 + b/S_1}{\sigma \sqrt{\sum_{i=2}^d (S_i/S_1)^2}} \right) \right]. \tag{50}$$

For perfect generalization, we must have $b/S_1 = -\mu$ and $S_1 \gg S_i, \forall i > 1$, which can only be achieved up to a certain error, determined by the max-margin solution.

In Fig. 16, we demonstrate numerically that while the limiting generalization accuracy is smaller than one for $r < 1$, grokking in the sense of delayed generalization and a non-monotonic test loss is still present.

### I.1. Derivation of Eq. (50)

Starting from the explicit expression for the accuracy

$$\mathcal{A} = \frac{1}{N_s} \sum_{i=1}^{N_s} \left( y_i \theta(S \cdot x_i + b) + (1 - y_i)\theta(-S \cdot x_i - b) \right), \tag{51}$$

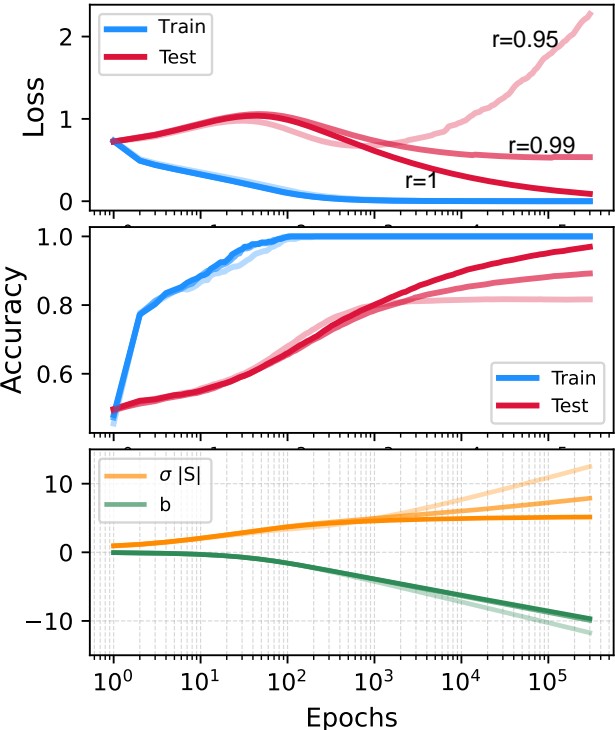

*Figure 16.* Grokking at $\lambda = 0.48$, for three different label-fractions: $r = 1$ (constant label), $r = 0.99$, and $r = 0.95$ (represented by three different opacities). For example, for $r = 0.95$, approximately $95\%$ of the input data would be assigned with the label $-1$ and $0.05$ with the label $1$. The rest of the parameters are the same as in Fig. 1.

and recalling that $\boldsymbol{x_i} \sim \mathcal{N}(0, \sigma \boldsymbol{I}_{d \times d})$, we get that the generalization accuracy is given by

$$\mathcal{A}_{\text{gen}} = \frac{1}{\sqrt{\det(2\pi\Sigma)}} \int d^d \boldsymbol{x} e^{-\frac{1}{2} \boldsymbol{x}^T \Sigma^{-1} \boldsymbol{x}} \left[ y(\boldsymbol{x}) \theta(S \cdot \boldsymbol{x} + b) + (1 - y(\boldsymbol{x})) \theta(-S \cdot \boldsymbol{x} - b) \right]. \tag{52}$$

Plugging the label to be a function of the first coordinate

$$y(\boldsymbol{x}) = \begin{cases} 0, & \boldsymbol{x}_1 < \mu \\ 1, & \text{else} \end{cases}, \tag{53}$$

where $\mu = Q(r)$ is the threshold ($Q$ being the Gaussian quantile function as discussed in the main text), we get that

$$\mathcal{A}_{\text{gen}} = \frac{1}{\sqrt{(2\pi\sigma^2)^{d-1}}} \int \prod_{j=2}^{d} dx_j e^{-\frac{1}{2\sigma^2} \sum_{i=2}^{d} x_i^2} \frac{1}{\sqrt{2\pi\sigma^2}} \left[ \int_{-\infty}^{\mu} dx_1 e^{-\frac{x_1^2}{2\sigma^2}} \theta(-S_1 x_1 - \sum_{i=2}^{d} S_i x_i - b) \right. \tag{54}$$

$$\left. + \int_{\mu}^{\infty} dx_1 e^{-\frac{x_1^2}{2\sigma}} \theta(S_1 x_1 + \sum_{i=2}^{d} S_i x_i + b) \right]. \tag{55}$$

Setting $y = \sum_{i=2}^{d} S_i x_i$, we note that $y \sim \mathcal{N}(0, \sigma_y)$, where $\sigma_y = \sigma \sqrt{\sum_{i=2}^{d} S_i^2} = \sigma \left( \|\boldsymbol{S}\| - S_1^2 \right)$. Therefore,

$$\mathcal{A}_{\text{gen}} = \frac{1}{\sqrt{2\pi\sigma_y^2}} \int_{-\infty}^{\infty} dy e^{-\frac{y^2}{2\sigma_y^2}} \frac{1}{\sqrt{2\pi\sigma^2}} \left[ \int_{-\infty}^{\mu} dx_1 e^{-\frac{1}{2\sigma^2} x_1^2} \theta(-S_1 x_1 - y - b) + \int_{\mu}^{\infty} dx_1 e^{-\frac{1}{2\sigma} x_1^2} \theta(S_1 x_1 + y + b) \right]. \tag{56}$$

Taking explicitly the integral over $y$ of the two terms and simplifying the result, we will finally get

$$\mathcal{A}_{\text{gen}} = \frac{1}{2} \left[ 1 + \frac{1}{\sqrt{2\pi\sigma^2}} \int_{-\infty}^{\infty} dx_1 e^{-\frac{1}{2\sigma^2} x_1^2} \text{sign}(x_1 - \mu) \text{erf}\left( \frac{1}{\sqrt{2}} \frac{S_1 x_1 + b}{\sigma\sqrt{||S||^2 - S_1^2}} \right) \right]. \tag{57}$$

## J. Direct calculation of the late time behavior of $b, S$ in the separable regime

Here we present a direct calculation for $\boldsymbol{S}(t) \equiv \|\boldsymbol{S}(t)\| \, \hat{\boldsymbol{S}}(t)$, $\boldsymbol{b}(t)$ at late times. First, we will work in the conformal time $\tau = \int_0^t \beta(t') dt'$. As discussed in the main text, when the data is separable we know that in the late time limit $\hat{\boldsymbol{S}}$ goes to a certain direction and $\|\boldsymbol{S}\| \to \infty$. Therefore, only the points with maximum $\boldsymbol{S}^T x_m$ will contribute to the sum in the large

$$\frac{1}{N} \sum_i e^{\boldsymbol{S}^T x_i} \approx \frac{D}{N} e^{\boldsymbol{S}^T x_m}, \tag{58}$$

where $\boldsymbol{S}^T x_m$ is the maximum value that is obtained, $D$ is their degeneracy. We note that $\boldsymbol{S}^T x_i$ are all negative, so the maximum are just the points which are closest to zero, so these are exactly the support vectors. To continue, for $\tau \to \infty$ we denote

$$\boldsymbol{S} \sim E f(\tau) \hat{\boldsymbol{S}}, \tag{59}$$

where $E$ is some constant, and $f(\tau)$ is some function of $t$. Without loss of generality, and for compatibility with the results of the main text, we will define $E$ by the equation $E \equiv -\frac{1}{\hat{\boldsymbol{S}}^T x_m}$. We can now find $f(\tau)$ explicitly using the following arguments: We know that

$$\frac{\partial \|\boldsymbol{S}\|}{\partial \tau} = \frac{\boldsymbol{S}^T}{\|\boldsymbol{S}\|} \frac{\partial \boldsymbol{S}}{\partial \tau} = -\eta \frac{1}{N} \sum_i e^{\boldsymbol{S}^T x_i} \frac{\boldsymbol{S}^T}{\|\boldsymbol{S}\|} x_i. \tag{60}$$

Using the approximation and comparing with $\frac{\partial \|\boldsymbol{S}\|}{\partial \tau} = E f'(\tau)$, we get that

$$\eta \frac{D}{N} \frac{1}{E} e^{-f(\tau)} = E f'(\tau). \tag{61}$$

Solving for $f(\tau)$, we get

$$f(\tau) = \log \left[ \eta \frac{D}{N} \frac{1}{E^2} \tau + C_1 \right], \tag{62}$$

where $C_1$ is some constant. Therefore, we get

$$\frac{\partial b}{\partial \tau} \approx -\eta \frac{D}{N} \frac{1}{\eta \frac{D}{N} \frac{1}{E^2} \tau + C_1}. \tag{63}$$

and taking the integral over $d\tau$ we get

$$\beta(\tau) \approx C_2 \left( \eta \frac{D}{N} \frac{1}{E^2} \tilde{t} + C_1 \right)^{-E^2}. \tag{64}$$

Recalling that $\frac{\partial \tau}{\partial t} = e^b$, we can integrate to find:

$$\tau(t) = \frac{1}{\eta \frac{D}{N} \frac{1}{E^2}} \left( \eta \frac{D}{N} \frac{E^2+1}{E^2} \right)^{\frac{1}{E^2+1}} [C_2 t + C_3]^{\frac{1}{E^2+1}} - \frac{1}{\eta \frac{D}{N} \frac{1}{E^2}} C_1. \tag{65}$$

Using $b = \log(\frac{\partial \tau}{\partial t})$, we get for long times that

$$b(t) \approx -\frac{E^2}{E^2+1} \log(t). \tag{66}$$

Plugging also $\|\boldsymbol{S}\| \approx E f(\tau) \approx E \log(\tau)$, we can see that

$$\|\boldsymbol{S}(t)\| \approx \frac{E}{E^2+1} \log(t). \tag{67}$$

Recalling that $E = \frac{1}{M}$, this verifies the result of the main text.

## K. Supplemental Details of Experiments

In this section, we will provide information regarding the experiments. The results of the left panels of Fig. 1 (and all of the results in Section App. F) can be easily obtained on a personal laptop. As for the heavier experiments which are presented in the right-most column of Fig. 1 (and in Fig. 7):

(1) Calculation of $\|\boldsymbol{S}_\infty\|$ and $\boldsymbol{S}^T x_i$ properties of the distribution. The setup is: $N = 2400$, $\sigma = 1$, and $d =$930, 990, 1050, 1086, 1110, 1134, 1152, 1158, 1164, 1170, 1173, 1176, 1179, 1182, 1185, 1188, 1191, averaged over 15000 different random realizations, and using the ADAM optimizer (any optimizer will work in the inseparable regime).

(2) Calculation of the margin. The setup is: $N = 2400$, $\sigma = 1$, and $d =$1230, 1260, 1290, 1320, 1350, 1380, 1410, 1440, 1470, 1500, 1530, 1560, averaged over 1000 realizations.

(3) The results of the right-middel and right-bottom panels of Fig. 1. The setup is: $N = 400$, and $d =$20, 40, 80, 100, 120, 140, 160, 168, 180, 188, 192, 196, 200, 208, 220, 228, 240, 260, 280, 300, 320, 360, averaged over 100 realizations.

