# OpenReview forum: "Grokking at the Edge of Linear Separability"
_ICML.cc/2025/Conference — ICML 2025 poster_

### Official Review · Reviewer_ZiSB · 2025-03-07

**Overall Recommendation:** 3

**Summary:**

The paper studies the grokking phenomenon and points out that grokking occurs near the critical point where data separability transitions. Specifically, the authors consider a simple logistic regression problem in the limit as the number of data points and the dimension of the model go to infinity. They show that there is a flat region in the loss landscape near the critical point, which causes delayed generalization.

## update after rebuttal
The authors' response was satisfactory, and I decided to maintain the score."

**Claims And Evidence:**

Claims are supported by theoretical results and / or numerical experiments.

**Essential References Not Discussed:**

Nothing in particular

**Experimental Designs Or Analyses:**

N/A

**Methods And Evaluation Criteria:**

N/A

**Other Comments Or Suggestions:**

Nothing in particular

**Other Strengths And Weaknesses:**

- strength
    - Grokking is an interesting phenomenon that has not been well understood.
    - The paper is well-organized and easy to follow.
    - The paper demonstrates the grokking phenomenon occurs in a simple logistic regression model.
- weakness
    - The paper focuses on the case where all inputs are assigned to the same label, which is very restrictive.
    Although the authors discuss the non-constant case in Section 5 but the condition is still restrictive.
    - Theoretical results is almost qualitative and qualitative results are provided for the simplified model.

**Questions For Authors:**

- Could you provide intuitive explanation on why grokking occurs even for $\sigma = 1$ when using Adam?
- What happens in the case where labels are not constant but the data is linearly separable?
- Which condition is essential for grokking to occur: linear separability or the labels being almost constant?

**Relation To Broader Scientific Literature:**

Grokking is an interesting phenomenon that has not been well understood while a lot of research has been done in recent years.
The paper offers a new perspective on grokking, which provides an interesting insight into the community.

**Theoretical Claims:**

I did not rigorously check the correctness of the proofs but the results seem to be reasonable.

---

> ### Author Rebuttal · Authors · 2025-03-31
>
> We greatly appreciate the reviewer’s positive evaluation and are pleased that they found our perspective on grokking to offer interesting insights. We have thoroughly addressed their concerns below. If they find our responses satisfactory, we hope they will be able to raise their confidence in accepting our work.
>
>
> **Weaknesses**
>
> - *"The constant label is restrictive”*: This appears to be the reviewer’s primary concern, so we will address it thoroughly. We believe this restriction is not as limiting as it may initially seem:
>
>     First, in Section 5, we demonstrate directly that even when the labels are discriminative, grokking can still be observed in the form of delayed generalization - there is a transition from memorization to generalization, albeit imperfect generalization.
>
>     Second, we believe that our model represents a *broader class* of models that can exhibit grokking for similar underlying reasons. Our focus on the specific case where all labels are the same was primarily motivated by its simplicity and the fact that it presents the phase transition in the cleanest manner.
>
>     To further support this claim, we will briefly describe a related sparse feature classification model that also demonstrates grokking behavior near the critical point with ***balanced labels***: Suppose that the first coordinate of the input, $x_1$, is distributed as a mixture of Gaussians with means $\pm \mu$ and variance $\sigma_1^2$, while all other coordinates $x_i$ for $i>1$ are drawn independently from $\mathcal{N}(0,\sigma^{2})$, as in our original setup. The label of each data point is determined by the sign of the first coordinate. The model is again logistic regression, but with *no bias term*. By choosing the ratio $\sigma/\sigma_1$ to be sufficiently small, one can observe arbitrarily large grokking time provided we are near the same critical point as our original model, at $\lambda=1/2$. The mechanism behind this behavior is essentially the same as described in the paper: the model initially converges to a memorizing solution (i.e., learning a random separating hyperplane) and only later transitions to the generalizing solution. A numerical demonstration of grokking for this model is included here: https://imgur.com/a/4BuMnls.
>
>     Please let us know if you would like additional analytical or numerical results regarding this model — especially if you believe it could affect your evaluation of the paper. Regardless, we will add to the appendix a brief discussion of this model.
>
> - *“Rigor of the results”*: We believe that our results are rigorous, note that some complementary proofs are left to the appendices. If any specific issue needs further clarification, please let us know and we will gladly address it.
>
>
> **Questions**
>
> - *“Why Adam works for $\sigma=1$*: We thank the reviewer for this interesting question. First, we will note that numerically one could see the exact same grokking behavior (for $\sigma=1$) not only for Adam, but also for much simpler optimizers such as SignGD (to which Adam behaves similarly, after some time, due to its ***adaptive*** nature). Therefore, we need to gain intuition for why $|S|$ grows faster than $b$ in SignGD, leading to grokking.
> We have $\frac{\partial|S|}{\partial t}=\frac{S}{|S|}\cdot\frac{\partial S}{\partial t}=|\frac{\partial S}{\partial t}|\cos(\alpha)$, where $\alpha$ is the angle between $\frac{\partial S}{\partial t}$ and $S$. Noting that $\frac{\partial S}{\partial t}$ is a vector of $\pm \eta$, we have $|\frac{\partial S}{\partial t}|=\eta\sqrt{d}$, where $d$ is the dimension. In the separable case (or on the verge of being separable), after some time the direction of $S$ saturates so we expect that $\alpha$ would be small. We thus get that $\frac{\partial|S|}{\partial t}\approx\eta\sqrt{d}\gg\eta=\frac{\partial b}{\partial t}$. To sum up, this happens due to the adaptive optimizer nature and the high-dimensionality of the model.
> - *Linearly separable data with discriminative labels*: For $\lambda<1/2$ but using discriminative labels, the model will exhibit late generalization even though the final accuracy differs from 1. For $\lambda>1/2$, similar to the constant-label case, the dynamics will lead us only toward the memorization solution, making the accuracy level stay at values close to its original value of $\approx 1/2$.
> - *“Which condition is essential for grokking to occur: linear separability or the labels being almost constant?”*. In our specific model, being close to the critical point with discriminative labels will result in delayed generalization but not necessarily full generalization. However, more generally, ***the proximity to the critical point*** (separability) is the essential point and ***not*** the constant labels: the grokking in the balanced-label model that was presented above is a clear demonstration of this fact.

---

### Official Review · Reviewer_DzGq · 2025-03-11

**Overall Recommendation:** 3

**Summary:**

The paper develops a minimal setup under the binary logistic classification task to theoretically characterize when and how grokking occurs. They provide both empirical and analytical insights into the mechanism of grokking. The theory utilizes past work on the implicit bias of gradient descent. The paper demonstrates that when the training data is highly unbalanced and on the verge of being linearly separable (from the origin), logistic regression can exhibit grokking.

**Claims And Evidence:**

All of the claims made in the paper are well supported by theoretical and empirical evidence.

**Essential References Not Discussed:**

I am not an expert in this area but I believe the related works section is quite thorough.

**Experimental Designs Or Analyses:**

Yes all the experiments are sound.

**Methods And Evaluation Criteria:**

Yes the proposed methods and evaluation criteria make sense for this application.

**Other Comments Or Suggestions:**

#### Minor:
- In Equation 3 it is a bit confusing to use the variable $y$ as a r.v. since the labels $y_i$ are just $-1$ for all $i$.
- Perhaps it would make more sense to state Prop. 3.1 as $b/\sigma \|\mathbf{S}\| \to -\infty$ to make the connection to Fig. 1 more clear. Or update Fig. 1 to plot $\|\mathbf{S}\|$ instead of $\sigma\|\mathbf{S}\|$.
- In the paragraph after the proof of Proposition 3.1 it should be clear that sub-optimal generalization implies
$c<\lim_{t \to \infty} \mathcal{A}_{\text{gen}}(\mathbf{S}(t),b(t))<1$ it needs to be lower bounded by a constant. Only having an upper bound of 1 does not indicate that it will not generalize. This is made clear in Proposition 3.2 but should be included in this section too.

**Other Strengths And Weaknesses:**

Overall the paper is quite easy to read and the ideas are explained very clearly. The only weakness is that the setting is extremely simple and it is unclear how we can connect this to even single hidden layer neural networks. Nonetheless, I believe this is a good first step and should not be used as a basis to reject the paper.

**Questions For Authors:**

1. One major concern I have about the paper is how it differs from other works which shown analytically that grokking can occur in simple models. For example: https://arxiv.org/abs/2310.16441. How do the results in this work differ? Does this work provide a better characterization of how the data influences whether or not grokking will occur?
3. I'm confused about the example in Section 4. On line 352 and 353. If $\lambda > 1/2$ then the data IS separable (from the origin) because both $x_1$ and $x_2$ are negative. In contrast, if $\lambda < 1/2$ then $x_1$ is negative and $x_2$ is positive so the data is NOT linearly separable from the origin. Is there a typo here? Same with line 369/370, isn't $\lambda > 1/2$ the separable case?

**Relation To Broader Scientific Literature:**

I actually do not understand how this fits into the broader scientific literature. Many of the paper mentioned in the related works section seem to be studying grokking looking at models that are amenable to analysis. What new insight does this model/setup provide?

**Theoretical Claims:**

I did not check the proofs carefully, but all the theoretical statements seem to be correct.

---

> ### Author Rebuttal · Authors · 2025-03-31
>
> We thank the reviewer for their detailed and valuable feedback.
>
> The reviewer's main concern is the simplicity of the setup and how our results could be generalized to other models (a point also raised by some of the other reviewers). However, the reviewer also noted that *“Nonetheless, I believe this is a good first step and should not be used as a basis to reject the paper”*. We believe that we have thoroughly addressed the remaining concerns below (please let us know if you think otherwise). In light of this, we hope the reviewer will consider raising their score. We will now address each point in detail.
>
> - **Relation to broader literature:**
>
>     *Novelty with respect to other analytically tractable models* - We apologize for not clarifying the difference between our work and previous works. Concretely, the papers cited in the *Related works* section which pertain to solvable models **do not** fully analytically solve the dynamics of the models under investigation and do not focus on ***criticality*** as a key aspect of the work. Typically, the results are semi-analytical (using DMFT or related techniques), with the only exception being Levi et al. (2023), which we will address in details below. We will revise this section to include this explanation in the final version.
>
>    In contrast to previous works, our model, being simple enough for analytical investigation, allows us to link the fundamental cause of grokking to the existence of ***critical points***. While we cannot yet prove this rigorously, we conjecture that grokking is intimately related to such critical points in settings beyond simple logistic regression. Most other grokking models are more complicated, making the underlying mechanism more elusive. For example, up until recently it was believed that feature learning and weight decay are necessary conditions for grokking, and our model shows this is not the case.
>
> - **Weaknesses:**
>
>     *The simplicity of the model*:  Rather than attacking the problem  from the direction of the ongoing vast research on the canonical examples of grokking (i.e., modular arithmetic), we try to approach it from a different direction by first analyzing its simpler manifestations. As such, we see the simplicity of the model as an advantage rather than a limitation. Of course, the next step must be trying to relate other examples of grokking to the same insights regarding criticality obtained from the simple model. However, we believe that this is a solid starting point, and that the community would benefit from the paper as it stands. We are currently working on using the criticality approach to study other models and also relating it to double descent — these would be addressed separately in our future works.
>
> - **Questions:**
>
>     1. *Differences from Levi et al, 2023*: We thank the reviewer for highlighting this important point. We are very familiar with the mentioned paper. The main and most fundamental difference between the two works is that, while Levi et al observe delayed generalization near the critical point, their setup does not display grokking in the regular sense of a transition from a memorizing to a generalizing solution. Simply speaking, their test loss does ***not*** exhibit non-monotonicity.
>
>         This limits the scope of their work, since non-monotonicity in the test loss is a hallmark of most known grokking examples. In contrast, our model is the first clean minimal example of grokking that naturally presents grokking both in delayed generalization ***and*** a memorization-generalization transition (see for example Fig. 5 in the appendix, where the non-monotonicity is more apparent since the y-axis is not in log-scaled).
>
>         Another obvious difference is that our model deals with cross-entropy loss rather than MSE, and as such, the dynamics and convergence to a solution are vastly different.
>
>         *It is, however, interesting to note that their result is also closely tied to criticality*: Taking a bit of inspiration from phase transitions in physical systems, we suspect that these two models may belong to ***different*** “universality classes” which are classes of transitions that all behave in the same manner in the proximity of the critical point (e.g., have the same critical exponents). The fact that these universality classes are affected only by “general” properties of the model (for example, its symmetries) makes their identification important, as they imply that one could deduce properties of a complex system by studying a much simpler model, as long as it has the same fundamental properties. While further research is certainly needed to verify this, it is part of our motivation for investigating these simpler models.
>
>     2. *Typos in the example in Sec. 4*: Thank you for pointing these out, both are indeed typos that will be fixed.
>
> - **Comments and suggestions:**
>
>     We appreciate all three of your suggestions and will clarify these points in the next revision.

---

### Official Review · Reviewer_hzJV · 2025-03-13

**Overall Recommendation:** 3

**Summary:**

This paper considers Grokking phenomenon on a simple binary logistic regression model. The authors consider gradient descent (GD) for solving a binary logistic regression problem and analyze the relationship between separability, generalization and overfitting.

In particular, they consider the case when the input data is generated from Gaussian distribution with labels being the same and the optimizer is gradient descent, what the training dynamics of certain quantities, such as generalization loss and accuracy, will behave under different conditions. They first prove the generalization is equivalent to separability, and then connect the separability with fixed $\lambda = d/N$ as $N, d\rightarrow \infty$ (where $N, d$ are number of data points and dimension) and reveal the grokking happens near $\lambda = 1/2$.

Then they discuss Grokking in discriminative labeling case (non-constant labels), and finally wrap up the paper by some discussions on limitations and future work.

**Claims And Evidence:**

The claims are mostly for a simple binary classification problem with toy settings, and they are well supported by the proof.

**Essential References Not Discussed:**

N/A

**Ethical Review Concerns:**

No concerns

**Experimental Designs Or Analyses:**

The experimental designs are sound.

The proposed methods are mainly for simple models from a theoretical perspective, and thus empirical verification can be focused on simple settings.

**Methods And Evaluation Criteria:**

The proposed methods are mainly some theoretical analyses of a simple model, and the evaluation criteria make sense.

However, it is unclear how to extend the analyses to more complex deep learning models, and in the paper there is no empirical investigation on this either.

**Other Comments Or Suggestions:**

1. Since the proof is mainly based on gradient flow, it might be better if the authors explicitly state it in Propositions? For example Prop 3.2 does not mention this, although they briefly mention it in Proof of 3.2.1 and 3.2.2. Discrete dynamical systems (gradient descent) can be quite different from continuous dynamical systems (gradient flow).

2. It might help if the authors can provide some definitions of certain terms in a separate and formal Def. For example having a separate Def for 'perfect generalization' might be better.

**Other Strengths And Weaknesses:**

Stengths:

1. The authors provide solid theoretical analyses of Grokking phenomenon on a simple linear model (Gaussian input, constant labels, and binary logistic regression) with gradient descent training dynamics.

Weaknesses:

1. This paper only considers simple models and the theory might not be extended to complex neural networks easily. As also mentioned by the authors in Section 6, they did not provide results on nonlinear logistic regression or even more complex deep learning models. The analyses are standard in theory community.

2. The experiments are mostly done under simple settings. It would be better if the authors can provide experimental results on some non-linear models that behave similar as what the theory suggests (e.g., different behavior when changing $\lambda$).

**Questions For Authors:**

1. Part of the proof is based on the fact that gradient descent dynamics can be analyzed through gradient flow. I am curious if it is possible to extend the current techniques to large learning rate settings, and thus connect the current framework to other settings like edge of stability and catapults.

2. The discussion in Section 5 is good. I am wondering if it is possible to extend the current proof to settings with non-constant labels. If not what would be the main challenges.

**Relation To Broader Scientific Literature:**

It is closely related to the machine learning theory community. The Grokking phenomenon itself is important, and is also related to edge of stability and catapults mechanism when learning rates are large.

**Theoretical Claims:**

I checked the correctness of part of the proofs, which are standard techniques such as gradient flow, inequalities, etc.

---

> ### Author Rebuttal · Authors · 2025-03-31
>
> We greatly appreciate the reviewer’s thoughtful feedback and positive appraisal of our work. We hope that by addressing the reviewer’s main concerns, they will be able to raise their confidence in accepting our work.
>
> **Weaknesses:**
>
> 1. *Limitations of simple models* - The reviewer's main concern is the applicability of our setting to more complex models. Our main claim is that our model contains interesting insights regarding the relationship between grokking and ***criticality***. In fact, the value of our model lies in its simplicity and the fact that it is analytically tractable, allowing us to ***isolate*** the mechanisms that cause grokking which are significantly harder to disentangle in more complex settings.
>
>
>     Naturally, the next step will be to extend our results to other models. A characterization of the exact relation between criticality and dramatic phenomena, including both grokking and double descent in complex settings, is currently a work in progress, and will appear in our future work. Nevertheless, we believe the community will benefit from the paper in its present form: please see also the arguments at the end of our response to Reviewer fNB5 (starting from: “To sum up, while our paper...”).
>
> 2. *Experimental results for non-linear models*: We agree that finding nonlinear models where our criticality results still hold would be interesting. However, we believe that extending our criticality analysis to general, nonlinear models warrants a separate work. The goal of this paper is to show a first, solvable model where the key features of grokking are clearly manifest (non-monotonic test loss and delayed generalization), in isolation from other mechanisms, such as weight decay or feature learning. Nonetheless, there are already many works on non-linear models acknowledging that the ratio of samples to dimensions is a crucial factor for grokking. See, for example, Appendix C, where we discuss the relation to canonical examples.
>
> **Comments and suggestions:**
>
> - Suggestions 1, 2: We thank the reviewer for both of these useful suggestions — we will incorporate these changes into the revised version.
>
> **Questions:**
>
> 1. *“Part of the proof is based on the fact that gradient descent dynamics...”*: Thank you for bringing up this interesting point. Empirically, we found that in our model, the maximum value that allows generalization is approximately $\eta = 1$, and that below this threshold our gradient-flow calculations fit quite well. Nonetheless, it could be interesting to investigate the effects of large learning rates and how they interact with the grokking phenomenon in this model.
>
> 2. *“The discussion in Section 5 is good... extend to settings with non-constant labels”*: It is indeed possible to extend our results. As is stated in Section 5, for non-constant labels, grokking in the sense of delayed generalization ***is*** observed, but not in the sense of perfect generalization. However, to demonstrate that grokking to perfect generalization does not require constant labels, we will present a ***balanced-label model that exhibits grokking***. Suppose the first coordinate $x_1$ is drawn from a Gaussian mixture with means $\pm \mu$ and variance $\sigma_1^2$, while the remaining coordinates $x_{i>1}$ are independently drawn from $\mathcal{N}(0,\sigma^2)$. Labels are determined by $\mathrm{sign}(x_1)$. Choosing a sufficiently small ratio $\sigma/\sigma_1$ yields arbitrarily large grokking times near the critical point $\lambda = 1/2$. Numerical results demonstrating grokking in this model can be found here: https://imgur.com/a/4BuMnls. Note that this also proves that the crucial point for grokking is the proximity to the critical point and not the constant labels, which is a good point raised by Reviewer ZiSB.
>
>
>
> Finally, we wish to highlight that our results hold not only for Gaussian inputs but for almost any data distribution (see Appendix H).

---

> > ### Comment · Reviewer_hzJV · 2025-04-04
> >
> > Thank you for your rebuttal. I would like to keep my score, and I encourage the authors to explore the extension of their theory to nonlinear models.

---

### Official Review · Reviewer_fNB5 · 2025-03-13

**Overall Recommendation:** 3

**Summary:**

This work investigates the grokking phenomenon where an increase in test performance is significantly delayed behind achieving perfect training performance. It primarily considered the highly simplified case of a linear model with constant labels, which is analyzed theoretically. In particular, the grokking effect is linked to the transition in $\lambda := \frac{d}{N}$ from the non-separable case (i.e. the data cannot be divided by a hyperplane from the origin) to the separable case (where dimensions increase or dataset size decreases such that this hyperplane exists). In the separable case, an overfitting solution will be found; in the non-separable case, a good solution will be found; but at the critical point between the two, a good solution will be found, but the time delay (in terms of gradient descent steps) can be arbitrarily long. These results are intuitively illustrated using an example with just two samples. Finally, in Section 5, some consideration given to the discriminative case where the labels are non-constant.

########## Update after rebuttal ##########

I thank the authors for their response. I have read their rebuttal in addition to the discussion with the other reviewers. I agree that the follow-up work discussed may help to extend the slightly limited scope of these findings. Despite the limitations discussed, I still lean towards acceptance and affirm my initial score.

**Claims And Evidence:**

I generally found the paper clear in its presentation, with the theoretical and experimental aspects convincing in their (specific) claims.

My main critique of this work is with respect to its broader argument that "the main takeaway from our setup is that grokking happens
near critical points." I agree that this claim is essential as, without connecting the highly simplified setting studied to standard grokking settings, the significance of this work would be extremely limited. However, I am not convinced that sufficient evidence (either experimental, theoretical, or through discussion) has been presented to defend this claim.

I would suggest that this claim would be far more convincing if (1) some concrete efforts were made to link the papers explanation to more realistic settings (particularly if some approaches to this extension could be implemented); and (2) if a more detailed reconciliation of the claims of this paper were made to the existing literature on grokking.

Regarding (2), one obvious counter example is that of existing cases of grokking in the deep learning literature in which grokking is reported even as the values of $N$ or $d$ are varied. I appreciate that Appendix C already relates to the original Power et al. work, but there are several other works that would also require reconciliation for the authors' proposed explanation to hold in general. For example, Figure 6 of [1] appears to imply that grokking occurs as dataset size varies. [2] argues that "being in the 'goldilocks zone for data set size' is necessary but not sufficient to see grokking" and argues for the relevance of alignment between features and the target function.  Several experiments in [3] relate dataset size to grokking. While I do accept that these works may be consistent with the claims of this work, if this is the case, that reconciliation must be made explicitly (see Section 9 of [2] for an example of this). Relatedly, it would appear that since the theory provided is closely tied to the norm of the coefficients, the authors could relate their findings to the relevance of weight decay/L2 regularization, which has been discussed in several works as an important aspect of grokking e.g. [4].


[1] Thilak, Vimal, et al. "The slingshot mechanism: An empirical study of adaptive optimizers and the grokking phenomenon." arXiv preprint arXiv:2206.04817 (2022).

[2] Kumar, Tanishq, et al. "Grokking as the transition from lazy to rich training dynamics." The Twelfth International Conference on Learning Representations.

[3] Liu, Ziming, et al. "Towards understanding grokking: An effective theory of representation learning." Advances in Neural Information Processing Systems 35 (2022): 34651-34663.

[4] Liu, Ziming, Eric J. Michaud, and Max Tegmark. "Omnigrok: Grokking Beyond Algorithmic Data." The Eleventh International Conference on Learning Representations.

**Essential References Not Discussed:**

See the previous section.

**Experimental Designs Or Analyses:**

Yes, all of the experimental results are numerical verifications that appear appropriate, given the theoretical setting.

**Methods And Evaluation Criteria:**

Given the highly simplified setting, the methods and general approach taken in this paper are suitable.

**Other Comments Or Suggestions:**

Minor:

*L408 should have no parentheses around (Schaeffer et al., 2023).

**Other Strengths And Weaknesses:**

Strengths:

* I found the exposition of this paper to be quite good in general.
* The figures were well made, clear, and were helpful in illustrating the points they intended to make.
* The writing was quite clear throughout, including when explaining the theoretical aspects of the work.
* Efforts were made to provide intuition around the results, which should make the paper somewhat more accessible to a general audience.
* The paper followed a clear logical flow.

**Questions For Authors:**

The results in this work hold "for any data distribution that is symmetric around the origin". Could the authors clarify if this implies that these results, therefore, apply for _any_ data distribution once the input distribution is standardized appropriately as we might do in practice?

**Relation To Broader Scientific Literature:**

As previously mentioned, this paper would have been greatly strengthened by efforts to verify its relevance to practical settings. While a complete explanation in all settings is not necessary, some concrete steps could certainly have been taken. For example, the aforementioned work of [2] uses a linear approximation to study the lazy regime and links grokking to a transition from lazy to rich training dynamics. Inutiively, some sort of separation of feature learning and a final layers classification would seem complimentary to this work. Similarly, [5] find a (piecewise) linear approximation of a neural network can capture the grokking effect which may offer a path towards extending the theory to full neural networks. It would seem that these linearized versions of neural networks would make a good starting point for relating this work to more practical settings, even if that link is primarily empirical.

Also, given that this work links grokking to the interpolation threshold as determined by $\lambda$ I think it would be valuable to provide even a brief connection with some of the work in the double descent literature. One particularly relevant example would be [6] where the same quantity is studied ($\gamma$ in that work). Several works have previously attempted to connect the two phenomena (e.g., [7,8]), and it would appear that there are some valuable connections to be made.




[2] Kumar, Tanishq, et al. "Grokking as the transition from lazy to rich training dynamics." The Twelfth International Conference on Learning Representations.

[5] Jeffares, Alan, Alicia Curth, and Mihaela van der Schaar. "Deep learning through a telescoping lens: A simple model provides empirical insights on grokking, gradient boosting & beyond." Advances in Neural Information Processing Systems 37 (2024): 123498-123533.

[6] Hastie, Trevor, et al. "Surprises in high-dimensional ridgeless least squares interpolation." Annals of statistics 50.2 (2022): 949.

[7] Davies, Xander, Lauro Langosco, and David Krueger. "Unifying grokking and double descent." arXiv preprint arXiv:2303.06173 (2023).

[8] Huang, Yufei, et al. "Unified view of grokking, double descent and emergent abilities: A comprehensive study on algorithm task." First Conference on Language Modeling. 2024.

**Theoretical Claims:**

I did not carefully check the proofs of the theoretical claims that were provided in the appendix. However, much of the used theory was adapting relatively standard ideas (i.e. the linear seperability of a dataset, which is understood in the standard logistic regression setting) to the grokking setting with gradient flow and seemed intuitively correct. This, combined with the numerical verifications, indicates that the theoretical claims are reasonable.

---

> ### Author Rebuttal · Authors · 2025-03-31
>
> We thank the reviewer for their positive and useful feedback, and are glad that the reviewer tends towards accepting our paper.
>
> The reviewer's primary concern is regarding the applicability of our setting to other existing models of Grokking in the literature, which we address below, along with other issues/questions raised.
>
> - *“this claim would be far more convincing...”*: While we believe our paper stands on its own (see the arguments at the end), we agree that it could benefit from a more detailed reconciliation with existing models. The reviewer has provided some good examples and directions. We will try to address some of them here, while an extension of this discussion will be added as an appendix.
>
> - *“..one obvious counter example...”*: Our conclusion does not mean that grokking/delayed generalization will be observed only for one ratio of $d/N$, but rather for a range of values *near* the critical point. However, a ***divergence*** of the “grokking time” appears only at the critical point itself. Note that for $N \gg d$ grokking will not exist.
>
> - *“Figure 6 of [1] appears...”*: We would expect that the grokking time would be *smaller* for a smaller ratio of $d/N$, but such data does not appear in the figure. Also, Thilak et al. only states that slingshots (not grokking itself) happen - it is not clear if there is a full correspondence. Regardless, the slingshot mechanism itself *relies* on adaptive optimizers (and $\varepsilon$), so it is not an ideal choice for extending the discussion regarding criticality.
>
> - *“being in the 'goldilocks zone..'”*. This aligns well with our results. Notice that grokking will not be observed even for $\lambda=1/2$ in our case, unless $\sigma$ is large enough to attract the dynamics toward the memorizing solution before generalizing (see rightmost panel of Fig. 3 in our paper). For more complicated settings, it is likely that other parameters will need to be tuned in order to see the grokking.
> - *Relation to weight decay:* We thank the reviewer for raising this point: It has recently become more apparent that WD is not a necessary condition for grokking, although it can help observe it in certain scenarios, see for example [4]. Our results support this observation: Adding WD will allow us to see grokking to a certain extent even in the non-separable region. We will add a discussion on the effects of WD in our model in the revised manuscript.
>
> - *“..the aforementioned work of [2] uses ...”* and *“Similarly, [5] find...”*: Thanks for highlighting these models, which are good candidates to investigate the criticality under broader settings. We will consider addressing it in our upcoming work, but we believe this is beyond the scope of the current work.
>
> - *Relation to double descent:* Thanks for this important note. We strongly believe such a relation exists, and that our model is a good candidate for investigating it. We are working in this direction and will address it separately in a future work.
>
> - *Response to the question*: Yes. In fact, these results will hold even for non-symmetric distributions, as this may only change the critical value of $\lambda$.
>
> To sum up, while our paper indeed does not offer a concrete recipe for how these conclusions extend to other models in the literature, we still believe that the community would benefit from its publication for the following reasons:
>
> 1. It provides a novel and analytically solvable model that exhibits grokking. Being minimalist, it highlights the true relationship between grokking and criticality — something that is somewhat obscured in more complex models.
>
> 2. In contrast to many of the current beliefs, we show that neither a transition from lazy to feature learning, nor WD are ***necessary conditions*** for grokking. Instead, the existence of a critical point is the necessary and sufficient requirement. If feature learning is required for this transition, these can coincide.
>
> 3. There are strong hints that suggest grokking is related to a criticality [1-3]. Our results support this claim, while offering analytical insights to its mechanism (our approach is somewhat similar to [2], but has much more benefits, see our response to reviewer DzGq for more details).
>
> 4. We believe that our results could lay the groundwork for revealing the relation between grokking, criticality, and double descent. In fact, we are currently working on these two directions (the relation to criticality and to double descent), and we intend to address them separately in our upcoming works.
>
> [1] Rubin et al., Grokking as a first order phase transition in two layer networks, 2024.
>
> [2] Levi et al., Grokking in linear estimators – a solvable model that groks without understanding, 2023.
>
> [3] Zhu et al. Investigate the critical data size for language model training through the lens of grokking dynamics, 2024.
>
> [4] Prieto et al. Grokking at the Edge of Numerical Stability, 2025.

---

### Decision · Program_Chairs · 2025-05-01

**Decision:**

Accept (poster)

**Comment:**

The authors present a detailed analysis of how grokking occurs near the threshold where the number of samples is equal to the input dimension.  They provide a precise analysis of when a solution is merely memorizing vs when it generalizes for a linear regression problem with Gaussian inputs.  The reviewers were largely in support of the work, with multiple pointed suggestions for how to further improve the presentation of the material and its relation to prior work.  I recommend acceptance.